# Demystifying Reasoning Dynamics with Mutual Information: Thinking Tokens are Information Peaks in LLM Reasoning

**Chen Qian[1,2*], Dongrui Liu[2*], Haochen Wen[3], Zhen Bai[4], Yong Liu[1], Jing Shao[2†]**
[1] Gaoling School of Artificial Intelligence, Renmin University of China
[2] Shanghai Artificial Intelligence Laboratory
[3] University College London, University of London   [4] Dalian University of Technology
{qianchen2022,liuyonggsai}@ruc.edu.cn   {liudongrui,shaojing}@pjlab.org.cn

## Abstract

Large reasoning models (LRMs) have demonstrated impressive capabilities in complex problem-solving, yet their internal reasoning mechanisms remain poorly understood. In this paper, we investigate the reasoning trajectories of LRMs from an information-theoretic perspective. By tracking how mutual information (MI) between intermediate representations and the correct answer evolves during LRM reasoning, we observe an interesting *MI peaks* phenomenon: **the MI at specific generative steps exhibits a sudden and significant increase during LRM's reasoning process**. We theoretically analyze such phenomenon and show that as MI increases, the probability of model's prediction error decreases. Furthermore, **these *MI peaks* often correspond to tokens expressing reflection or transition, such as "Hmm", "Wait" and "Therefore,"** which we term as the *thinking tokens*. We then demonstrate that these *thinking tokens* are crucial for LRM's reasoning performance, while other tokens has minimal impacts. Building on these analyses, we propose two simple yet effective methods to improve LRM's reasoning performance, by delicately leveraging these *thinking tokens*. Overall, our work provides novel insights into the reasoning mechanisms of LRMs and offers practical ways to improve their reasoning capabilities. The code is available at `https://github.com/ChnQ/MI-Peaks`.

## 1 Introduction

The reasoning ability of large language models (LLMs) has emerged as one of their most powerful and crucial capabilities [53, 21, 22]. By explicitly thinking through a question before providing an answer and breaking down complex problems into multiple steps, LLMs have made impressive progress in complex reasoning tasks, such as mathematics, programming, and logical inference [28, 60, 45, 7]. Understanding and improving LLMs' reasoning ability represents a crucial pathway toward achieving Artificial General Intelligence (AGI) [56, 52, 43].

By undergoing reasoning-intensive training on foundational LLMs, recent large reasoning models (LRMs) such as OpenAI's o1 [22], DeepSeek's R1 [19], and QwQ [46] have demonstrated exceptional reasoning capabilities, significantly pushing the boundaries of complex problem-solving. However, despite recent advances, the mechanisms underlying these capabilities remain largely under-explored. The internal dynamics of the reasoning process, as well as the influence of each intermediate step on the final answer, are still largely a "black box." While some research in the field of trustworthy AI

---

* Equal contribution. This work was done during an internship at Shanghai Artificial Intelligence Laboratory, supervised by Dongrui Liu.    † Corresponding author

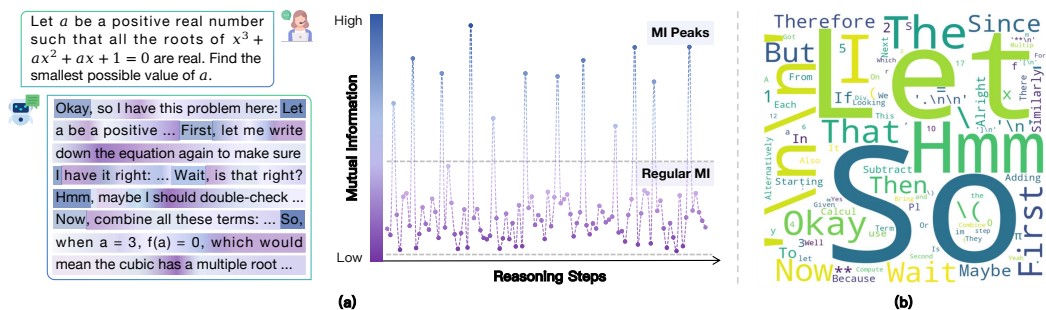

Figure 1: Illustration and analysis of the MI peaks phenomenon in LRM reasoning. (a) The **left** side shows an example of an LRM performing a multi-step reasoning task. To investigate the underlying reasoning mechanism, we compute the MI between the model's representation at each step and the golden answer. Interestingly, as shown on the **right** side, certain steps exhibit sudden and significant increases in MI, which we refer to the MI peaks phenomenon. (b) Token distribution at MI peaks. We further find that the tokens generated at these high-MI steps are often reflective or transitional expressions such as "So," "Hmm," and "Wait."

suggests the existence of "critical tokens" that directly impact the safety of the LLM's answers [65, 29, 36], a natural question arises: *are there critical reasoning steps or intermediate states that significantly affect the final results in the reasoning process of LRMs?*

In this paper, we explore this question from an information-theoretic [4, 27] perspective. Specifically, given a question, we dynamically calculate the mutual information (MI) between the LRM's representation at each step of reasoning process and the golden answer (*i.e.,* the ground-truth response), observing how the MI evolves. Interestingly, we find that **certain steps' representations exhibit a sudden and significant increase in MI with the golden answer**. As shown in Figure 1(a), these representations with MI peaks are sparse and occur non-uniformly throughout the reasoning process. This suggests that at certain crucial reasoning steps, LRMs' representation becomes highly informative about the correct answer. Naturally, this raises a question: *are these MI peaks potentially related to model's reasoning performance?* Theoretically, we provide preliminary insights into the MI peaks phenomenon, demonstrating that as the cumulative MI between the representations and the golden answer increases, the probability of LRM's wrong prediction lowers. Furthermore, our experiments show that the base models corresponding to these LRMs (*e.g.,* LLaMA-3.1-8B [17]), does not exhibit this MI Peaks phenomenon as clearly. These analyses suggest that the distinct MI peaks observed during LRM reasoning are potentially stemming from the reasoning-intensive training, and may hold a potential relationship with LRM's advanced reasoning abilities.

This naturally leads to the question: *what semantic roles do the representations at MI peaks play during reasoning?* Intriguingly, we find that **these representations with MI peaks predominantly correspond to tokens such as "Wait," "Hmm," "Therefore," "So," which typically express reflectiveness, self-correcting, or transitions**, as shown in Figure 1(b). Here, we refer to these tokens with MI peaks as "thinking tokens". Since these thinking tokens explicitly prompt the model to reflect and reason, and their representations carry enriched information with the golden answer, we hypothesize that these thinking tokens may play a critical role in the model's reasoning ability. To validate this hypothesis, we suppress the generation of these thinking tokens and observe how the model's reasoning performance changes. As shown in Figure 5, fully suppressing the generation of these thinking tokens significantly harms the model's reasoning performance, while randomly suppressing the same number of tokens has little impact. This indicates that these thinking tokens are indeed crucial to LRM's reasoning ability.

Finally, drawing insights from the above analyses, we propose to improve the reasoning performance of LRMs in two training-free ways. 1) By allowing the representations at MI Peaks to undergo multiple iterations within the model, we propose a method called Representation Recycling (RR). RR encourages the model to better exploit these informative representations. Experiments show that RR consistently improves the LRMs' reasoning performance across several benchmarks. For instance, it improves the accuracy of DeepSeek-R1-Distill-LLaMA-8B by 20% relatively on AIME24. 2) Motivated by our analysis of *thinking tokens*, we propose Thinking Token based Test-time Scaling (TTTS). That is, when additional token budget remains, we force the model to continue reasoning by begin with the *thinking tokens*. Experiments show that TTTS leads to steady performance

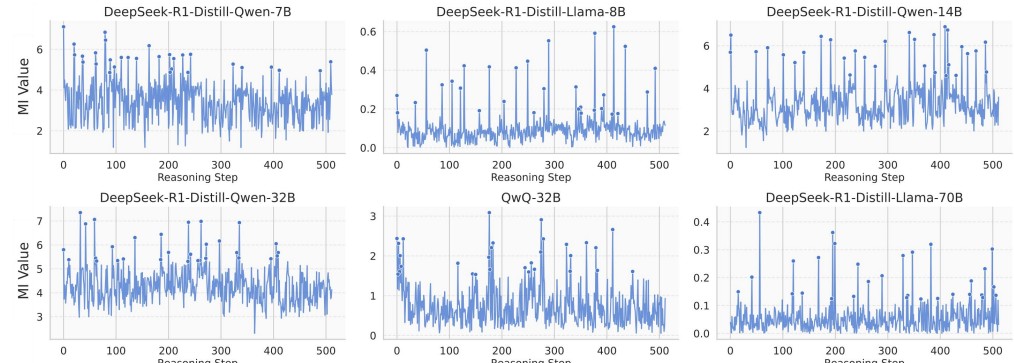

Figure 2: The evolution trajectories of MI between each step's representations and the golden answer during the reasoning process in LRMs.

improvements as the token budget increases compared to the original LRMs. These applications further demonstrate that our observations can offer new insights into enhancing the reasoning abilities of LRMs.

## 2 Emergence of MI Peaks in LRMs' Reasoning Trajectories

Despite the impressive reasoning capabilities demonstrated by recent LRMs such as DeepSeek's R1 series models [19] and Qwen's QwQ [46], the underlying mechanisms driving these capabilities remain poorly understood. In this section, we investigate the reasoning trajectories of LRMs from an information-theoretic perspective. We begin by introducing the notations and preliminaries (Section 2.1). In Section 2.2, we demonstrate the MI peaks phenomenon. We then provide theoretical insights into this phenomenon in Section 2.3. Finally, we examine whether similar patterns emerge in the corresponding non-reasoning LLMs of LRMs in Section 2.4.

### 2.1 Preliminaries

**Extracting representations in LRM generation process.** Given a data sample $s = (x, y)$, where $x$ is the input query and $y$ denotes the corresponding golden answer, which consists of both the intermediate chain-of-thought reasoning steps and the final solution. For a LLM $\mathcal{M}$, when prompted with $x$, it auto-regressively generates $\hat{y} = \{\hat{y}_1, \hat{y}_2, \ldots, \hat{y}_T\}$, where $T$ is the total number of tokens and $\hat{y}_t$ denotes the token produced at step $t$. To analyze the dynamic generation process, we collect the hidden representation corresponding to each generated token. Let $\mathcal{A}_i^l(\cdot)$ denote the representation extraction function that extracts the representation of the $i$-th token at layer $l$ of a LLM when given an input. For simplicity, we omit the superscripts and subscripts on $\mathcal{A}$. In this way, the representation corresponding to the $t$-th generated token is denoted by $\boldsymbol{h}_t = \mathcal{A}\big(\mathcal{M}(x, \hat{y}_{<t})\big)$, where $\hat{y}_{<t}$ denotes the subsequence of $\hat{y}$ before the $t$-th token. Similarly, we also extract the representation of the gold answer by feeding $y$ into the LLM, *e.g.*, $\boldsymbol{h}_y = \mathcal{A}\big(\mathcal{M}(y)\big)$.

**Estimating MI between each generated token and golden answer.** After extracting the representation, we then measure the MI between each generated token's representation $\boldsymbol{h}_t$ and the golden answer's representation $\boldsymbol{h}_y$, obtaining a MI sequence: $I[\boldsymbol{h}_1; \boldsymbol{h}_y], I[\boldsymbol{h}_2; \boldsymbol{h}_y], \ldots, I[\boldsymbol{h}_T; \boldsymbol{h}_y]$. In this way, we observe how MI evolves, thus analyze the reasoning dynamics during LLM's generation process. Specifically, we follow [32, 38, 13] to use the Hilbert-Schmidt Independence Criterion (HSIC) [18] to estimate MI [27, 35]. The formal definition of HSIC is stated in Definition 4, and we provide more implementation details in Appendix B.

**Definition 1** (Hilbert-Schmidt Independence Criterion (HSIC) [18])**.** HSIC *is the Hilbert-Schmidt norm of the cross-covariance operator between the distributions in Reproducing Kernel Hilbert Space (RKHS). Formally:*

$$
\begin{aligned}
\mathrm{HSIC}(X, Y) =& \mathbb{E}_{XYX'Y'}\left[k_X\left(X, X'\right) k_Y\left(Y, Y'\right)\right] + \mathbb{E}_{XX'}\left[k_X\left(X, X'\right)\right] \mathbb{E}_{YY'}\left[k_Y\left(Y, Y'\right)\right] \\
&-2\mathbb{E}_{XY}\left[\mathbb{E}_{X'}\left[k_X\left(X, X'\right)\right] \mathbb{E}_{Y'}\left[k_Y\left(Y, Y'\right)\right]\right],
\end{aligned}
$$

(1)

*where $X'$, $Y'$ are independent copies of $X$, $Y$, respectively, and $k_X$, $k_Y$ are kernel functions.*

Table 1: Statistical properties of MI peaks across different LRMs. Here, *#MI Peaks* and *#All Steps* refer to the number of MI peaks and the total number of reasoning steps, respectively. *Interval of MI Peaks* denotes the number of steps between two consecutive MI peaks.

| Model | #MI Peaks | #All Steps | Ratio of MI Peaks | Max Interval of MI Peaks | Min Interval of MI Peaks | Avg Interval of MI Peaks |
|---|---|---|---|---|---|---|
| DeepSeek-R1-Distill-Qwen-7B | 2.57 | 507.97 | 0.0051 | 152.67 | 52.74 | 87.38 |
| DeepSeek-R1-Distill-Llama-8B | 24.54 | 511.03 | 0.0480 | 69.37 | 6.65 | 27.84 |
| DeepSeek-R1-Distill-Qwen-14B | 18.30 | 510.09 | 0.0359 | 85.50 | 5.33 | 31.09 |
| DeepSeek-R1-Distill-Qwen-32B | 10.82 | 511.22 | 0.0212 | 138.07 | 19.35 | 59.30 |
| QwQ-32B | 5.41 | 489.80 | 0.0110 | 167.85 | 19.35 | 66.53 |
| DeepSeek-R1-Distill-Llama-70B | 16.60 | 512.00 | 0.0324 | 93.03 | 6.77 | 34.71 |

## 2.2 Investigating LRM's Reasoning Trajectories with MI

In this subsection, we track how the MI between each step's representation and the gold answer evolves, following the procedure in Section 2.1. Specifically, we conduct experiments on several popular LRMs of varying scales, including the DeepSeek-R1-Distill series [19] and QwQ-32B [46]. We use the training split of the MATH dataset [20], which comprises 12k competition-level mathematics problems, each accompanied by a detailed step-by-step solution.

**Certain steps exhibit sudden and significantly increases in MI during the reasoning process of LRMs.** Figure 2 shows the MI evolution trajectories for one data sample during LRMs generation[1]. Surprisingly, across all tested LRMs, we observe a consistent pattern: while most steps exhibit relatively low and stable MI values as reasoning proceeds, certain steps' MI suddenly and significantly increases. We refer to these steps with abrupt increase in MI as the *MI peaks*. Formally, we define MI peaks as follows:

**Definition 2** (MI Peak). *Given a MI sequence $\{m_t\}_{t=1}^T$, let $Q_1$, $Q_3$ denote the 25-th percentile (first quartile), and the 75-th percentile (third quartile) of the sequence, respectively. We then define* $\mathrm{IQR}(m) = Q_3 - Q_1$ *as the inter-quartile range. In this way, we identify the set of MI peaks as*

$$\mathcal{O} = \big\{ t : m_t > Q_3 + \tau \, \mathrm{IQR}(m) \big\},$$

*where $\tau$ is a scale factor. Empirically, we set $\tau$ to $1.5$ [48].*

**MI peaks are sparse and distribute non-uniformly throughout the total reasoning process.** As shown in Table 1, MI peaks occur quite sparsely in the reasoning processes of LRMs, accounting for no more than 5% of all reasoning steps. Notably, for DeepSeek-R1-Distill-Qwen-7B, the MI peak ratio is only 0.51%. Despite this sparsity, these MI peaks are scattered across the entire reasoning trajectory, as illustrated in Figure 2. Moreover, the interval statistics reported in Table 1 indicate that MI peaks do not occur at uniform intervals. Such a sparse and non-uniform distribution pattern suggests that MI peaks may emerge opportunistically at key moments during reasoning.

## 2.3 Theoretical Insights: Higher MI Leads to Tighter Bounds on Prediction Error

In Section 2.2, our empirical exploration reveals the emergence of MI peaks in LRMs' reasoning trajectories, indicates that certain representations encode substantially rich information about the gold answer. This raises a natural question: *would such pattern be potentially related to the LRM's reasoning performance?* In this subsection, we provide theoretical insights into this question, showing that higher MI between the representations and the gold answer yields tighter lower and upper bounds on the model's prediction error.

**Theorem 1.** *Consider a sequence of representations $\boldsymbol{h}_1, \boldsymbol{h}_2, \ldots, \boldsymbol{h}_T$ during an LLM's reasoning process, where $T$ denotes the number of total reasoning steps. Let $y$, $\hat{y}$ denote the golden answer and the LLM's prediction answer, respectively. Define $p_e = \mathrm{Pr}(\hat{y} \neq y)$ as the LLM's prediction error probability. Then the following inequality holds:*

$$p_e \geqslant \frac{1}{\log(|\mathcal{Y}| - 1)} \Big[ H(y) \; - \; \sum_{j=1}^{T} I\big(y; \boldsymbol{h}_j \mid \boldsymbol{h}_{<j}\big) \; - \; H_b(p_e) \Big], \tag{2}$$

---

[1]Results for more examples and more LRMs are reported in Appendix D.

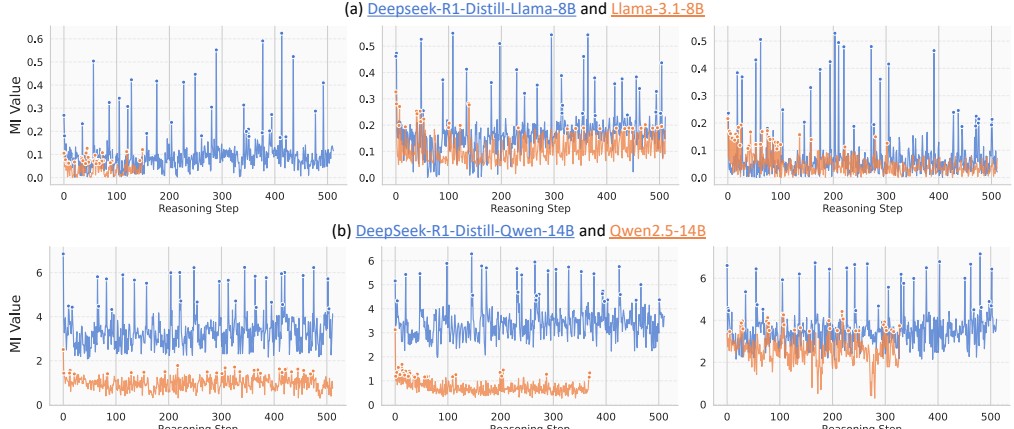

Figure 3: Comparison of MI trajectories between LRMs and their corresponding non-reasoning LLMs.

where $|\mathcal{Y}|$ is the size of the support of $y$, and $H_b(p_e)$ denote the binary entropy of $p_e$ that defined by

$$H_b(p_e) = -p_e \log p_e - (1 - p_e) \log(1 - p_e). \tag{3}$$

**Remark 1.** Theorem 1 establishes a lower bound on the LLM's prediction error $p_e$. Intuitively, it suggests that for an LLM to achieve a low error rate, its sequence of internal representations during generation should capture more information about the golden answer. In other words, higher MI throughout the generation trajectory may help lower model's minimal achievable error. Note that, this result be viewed as a modified application of *Fano's inequality* [11], adapted to step-wise reasoning by decomposing the mutual information along the trajectory.

**Theorem 2.** *Following the notations in Theorem 1, the following inequality holds:*

$$p_e \;\leqslant\; \frac{1}{2}\Big[H(y) \;-\; \sum_{j=1}^{T} I\big(y;\, \boldsymbol{h}_j \mid \boldsymbol{h}_{<j}\big)\Big]. \tag{4}$$

**Remark 2.** Theorem 2 provides an upper bound on the prediction error $p_e$, which complements the lower bound in Theorem 1. It demonstrates that a higher cumulative MI between the sequence of representations and the golden answer leads to a tighter upper bound on LLM's error probability.

**Remark 3.** In summary, Theorems 1 and 2 jointly suggest that, higher cumulative MI between representations during reasoning and the golden answer leads to a tighter upper and lower bounds on the model's error probability. In other words, the model is more likely to arrive at the correct answer. Notably, the presence of MI peaks can effectively increase this cumulative MI, thereby potentially helping LLMs to perform more accurate reasoning.

### 2.4 Will Non-reasoning LLMs also Exhibit the MI Peaks Phenomenon?

Since the MI Peaks phenomenon is commonly observed in LRMs, *would non-reasoning LLMs (i.e.,* foundation LLMs not specifically strengthened for complex reasoning, such as Llama-3.1-8B [17]) *also exhibit similar behavior?* To explore this question, we select the corresponding non-reasoning counterparts of the DeepSeek-R1-Distill series models and follow the workflow described in Section 2.1 to conduct experiments.

**Metrics.** To facilitate a quantitative comparison between LRMs and their corresponding base models in terms of the properties of MI sequence $\{m_t\}_{t=1}^{T}$ during reasoning, we adopt the following metrics: (1) *Mean*: $\bar{m} = \frac{1}{T} \sum_{i=1}^{T} m_i$; (2) *Standard deviation (Std)*: $\sigma_m = \sqrt{\frac{1}{T} \sum_{i=1}^{T} \big(m_i - \bar{m}\big)^2}$; (3) *AOM*: $\text{AOM} = \frac{1}{|\mathcal{O}|} \sum_{i \in \mathcal{O}} \frac{|m_i - \text{median}(m)|}{\text{IQR}(m)}$, where $\mathcal{O}$ is the set of MI peaks defined in Definition 2, $\text{median}(m)$ is the median of the sequence $\{m_t\}_{t=1}^{T}$. Specifically, *Mean* reflects the overall MI magnitude, while the *Std* and *AOM* capture the degree of MI fluctuation.

**Non-reasoning LLMs exhibit weaker and less pronounced MI peaks compared to LRMs.** As shown in Figure 3, while certain steps in non-reasoning LLMs' reasoning process do exhibit increased MI relative to the average, the increase is generally mild and lacks the sharp spikes observed in

Table 2: Statistical comparison of MI sequences between LRMs and their corresponding non-reasoning LLMs.

| Metric | Llama-3.1-8B | | Qwen2.5-Math-7B | | Qwen2.5-14B | | Qwen2.5-32B | | Llama-3.3-70B-Inst | |
|---|---|---|---|---|---|---|---|---|---|---|
| | Origin | Reasoning | Origin | Reasoning | Origin | Reasoning | Origin | Reasoning | Origin | Reasoning |
| Mean | 0.0863 | 0.1279 | 2.1971 | 3.3016 | 1.3128 | 3.3508 | 1.7669 | 4.0352 | 0.0400 | 0.0599 |
| Std | 0.0512 | 0.0707 | 0.8639 | 0.8936 | 0.4326 | 0.6703 | 0.5113 | 0.6036 | 0.0277 | 0.0484 |
| AOM | 3.3573 | 4.5176 | 2.6320 | 2.7541 | 2.6541 | 3.0820 | 2.5466 | 2.5998 | 2.4326 | 3.2866 |

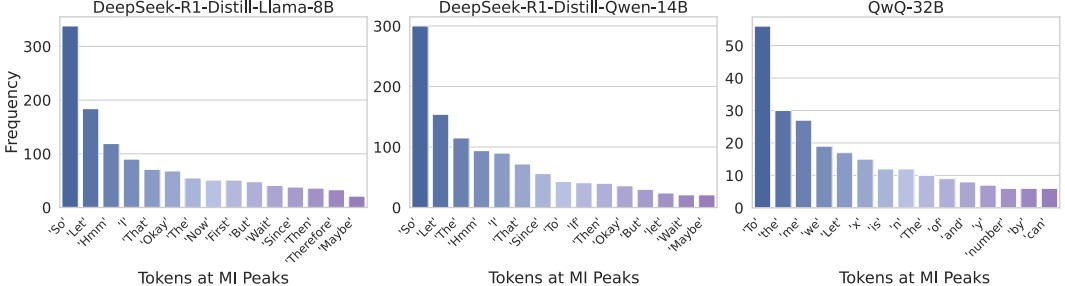

Figure 4: Frequency distribution of tokens at MI peaks.

their LRM counterparts. Quantitatively, this observation is further supported by the *Std* and *AOM* metrics reported in Table 2, which consistently indicate lower MI fluctuation and peak intensity in non-reasoning LLMs. These findings suggest that the MI peak pattern may emerges from complex reasoning enhanced training.

**The overall MI in non-reasoning LLMs during the reasoning process is lower than their corresponding LRMs.** Figure 3 and the *Mean* metric in Table 2 intuitively and quantitatively validate this observation, respectively. This indicates that after reasoning-intensive training, LRMs seems to fundamentally encode more information relevant to correct reasoning within their representations at each generation step. Furthermore, the presence of MI peaks in LRMs could contribute to raising the overall MI throughout the reasoning trajectory. These observations provide partial empirical support for the theoretical insights presented in Section 2.3, which indicate that higher MI between representations and the golden answer correlates with a greater likelihood of generating a correct response.

## 3 Thinking Tokens are Information Peaks in LLM Reasoning

In Section 2, we identify a distinctive phenomenon in LRMs' reasoning trajectories: the emergence of MI peaks. Then a natural follow-up question is: *what semantic information is encoded in the representations at these MI peaks?* In this section, we investigate this question from a token-level perspective. Specifically, in Section 3.1, we project the representations at MI peaks into the token space and analyze the characteristics of the corresponding tokens. Then in Section 3.2, we design experiments to assess the functional role of these tokens, demonstrating that they are crucial for LRM's reasoning performance, while other tokens have minimal impact.

### 3.1 Exploring MI Peak Representations in Token Space

**Projecting representations to token space.** To interpret the semantics of representations at MI peaks, we decode these specific representations into the token space using LLM's output head [50, 59, 15]. Specifically, for a representation $\boldsymbol{h}_t$, we first compute the corresponding token probability distribution, and then employ a greedy decoding strategy to extract the token with the highest probability:

$$\boldsymbol{p}_t = \text{Softmax}(\boldsymbol{W}_{\text{out}}\boldsymbol{h}_t + \boldsymbol{b}), \quad \hat{z}_t = \arg\max_{i \in \{1,...,V\}} [\boldsymbol{p}_t]_i, \tag{5}$$

where $\boldsymbol{W}_{\text{out}} \in \mathbb{R}^{V \times d}$ is the output projection matrix, $\boldsymbol{b} \in \mathbb{R}^V$ is the bias vector, and $V$ is the vocabulary size. We apply the above decoding procedure to all representations at MI peaks across the evaluation dataset. In this way, we analyze the empirical distribution over these decoding tokens, uncovering

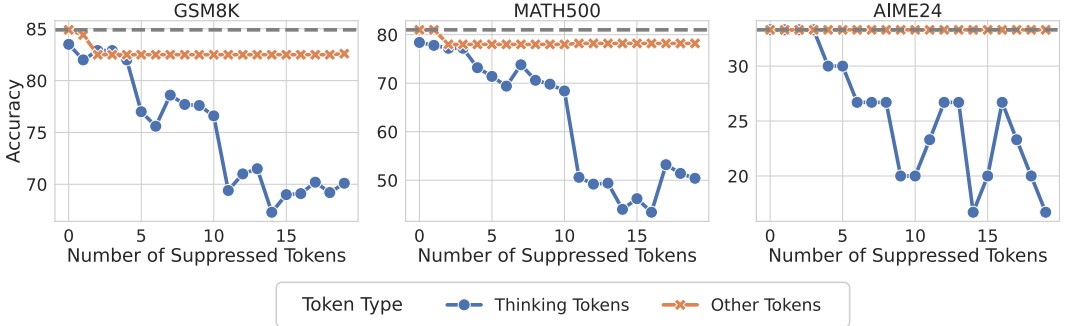

Figure 5: Impact of suppressing the generation of thinking tokens versus other tokens on LRMs' reasoning performance.

patterns about what types of semantic tokens tend to correspond to these high-MI representations. Specifically, we use the same models and dataset as described in Section 2.1 to conduct experiments. For each model, we aggregate all decoded tokens at MI peaks across the dataset, and then compute their frequency distribution for further analysis.

**The tokens that appear at MI peaks are mostly connective words that express self-reflection or transitions in LRM's reasoning process.** In Figure 4, we illustrate the top-30 tokens decoded at MI peaks in DeepSeek-R1-Distill-LLaMA-8B, DeepSeek-R1-Distill-Qwen-14B and QwQ[2]. Interestingly, we observe that the MI peak tokens in LRMs are predominantly logical markers and reflective expressions such as *"So"*, *"Hmm"*, and *"Wait"*, which are commonly associated with pause, thinking, or internal deliberation. Intuitively, tokens like *"Hmm"* and *"Wait"* often prompt the model to self-reflect, consider alternative reasoning paths, etc. For example, we randomly extract responses from LRMs where these tokens appear and observe the follow-up statements: "Wait, let me think differently. Let's denote...," "Hmm, so I must have made a mistake somewhere. Let me double-check my calculations. First, ..." This behavior aligns with prior work suggesting that such tokens can motivate to perform multi-step reasoning and improve answer accuracy [19]. We provide more discussions in Appendix C.

## 3.2 Tokens at MI Peaks are Critical to LRM's Reasoning Performance

Here, we refer to those decoded high-MI tokens in Section 3.1 as *thinking tokens*. These thinking tokens appear to play a dual role: (i) linguistically, they serve as discourse cues that encourage the model to think or reflect; and (ii) in hidden space, their corresponding representations contain high MI with the golden answer. Thus, we hypothesize that *these thinking tokens may be critical to model's final reasoning results*. In this subsection, we conduct experiments to validate this hypothesis.

**Suppressing the generation of thinking tokens significantly impairs the reasoning performance of LRMs, while suppressing other tokens has minimal effect.** To investigate the role of thinking tokens identified at MI peaks, we conduct a controlled intervention experiment. Specifically, during inference with LRMs, we suppress the generation of a certain number of thinking tokens by setting their generation probabilities to zero. As a comparison, we randomly suppress the same number of non-thinking token. In this way, we evaluate the model's performance on several math reasoning benchmarks under different numbers of suppression tokens. As shown in Figure 5, suppressing thinking tokens leads to a significant degradation in the model's reasoning performance, while suppressing non-thinking tokens has little to no effect (more discussions are provided in Appendix C). This indicates that the thinking tokens indeed play a critical role in LRMs' reasoning capabilities, providing empirical support for our previous hypothesis.

# 4 Applications: Leveraging MI Peaks to Improve LRM Reasoning

Drawing insights from our previous analyses, we propose two simple yet effective techniques to improve LRMs' reasoning performance. In Section 4.1, we introduce a method that reuses internal representations at MI peaks to allow the model to further exploit the information in latent space. In

---

[2]Results for the other models are provided in Appendix D.

Section 4.2, we incorporate the thinking tokens into a test-time scaling scenario to improve model's reasoning accuracy.

## 4.1 Recycling High-MI Representations During Inference

The MI Peaks phenomenon analyzed in Section 2.2 suggests that some representations in LRMs' reasoning process may encode particularly useful semantic information for reasoning. Motivated by this, we propose a simple technique named Representation Recycling (RR). Intuitively, RR feeds the representations at MI peaks back into the model, thereby allowing the model to process and exploit these representations more thoroughly.

**Method.** Recall that each layer in an LLM typically consists of a Transformer block [49]. Given an input, the forward computation flow through the layers of an LLM follows:

$$\boldsymbol{h}_\ell = \mathrm{TF}_\ell(\boldsymbol{h}_{\ell-1}), \quad \ell = 1, \ldots, L,$$

where $\boldsymbol{h}_\ell$ is the output representation of the $l$-th transformer block $\mathrm{TF}_\ell(\cdot)$, and $L$ is the total number of layers. To encourage deeper processing of a potentially important representation $\boldsymbol{h}_{\ell*}$ at layer $\ell^*$, we modify the forward computation by feeding it back into the same layer once more: $\boldsymbol{h}'_{\ell*} = \mathrm{TF}_{\ell*}(\boldsymbol{h}_{\ell*})$, instead of directly passing it to the next layer. Then, for layers $\ell > \ell^*$, we continue the forward pass as usual: $\boldsymbol{h}'_\ell = \mathrm{TF}_\ell(\boldsymbol{h}'_{\ell-1})$. In this way, the above "recycling" operation allows the model to reprocess the high-MI representations to further extract critical reasoning features.

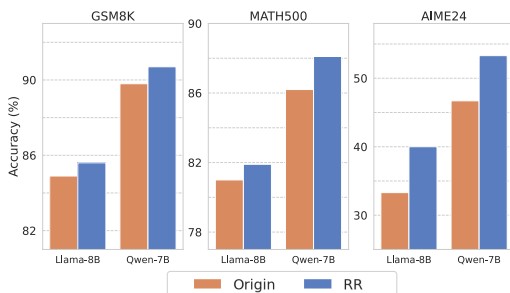

Figure 6: Reasoning performance of the original LRMs and our RR method across multiple math benchmarks.

**Experimental setup.** To evaluate RR's effectiveness, we conduct experiments on three mathematical reasoning benchmarks using DeepSeek-R1-Distill-Llama-8B and DeepSeek-R1-Distill-Qwen-7B. Since ground-truth answers are unavailable during inference, we first record the thinking tokens using the training set of MATH dataset (as introduced in Section 3.1), and then trigger RR whenever the model generates one of these thinking tokens. We empirically set $\ell^*$ to middle or high layers of the LLMs, since previous studies suggest that these layers tend to encode more semantically rich content [6, 64, 38].

**Results.** As shown in Figure 6, **RR consistently improves LRMs' reasoning performance across all benchmarks**. In particular, RR yields a notable performance improvement on the AIME24 dataset, which consists of challenging competition-level problems. This suggests that recycling the MI-peak representations could help LRMs further unlock and leverage their inherent reasoning potential, leading to better reasoning performance.

## 4.2 Test-Time Scaling with Thinking Tokens

With the diminishing returns of scaling laws in LLMs' training stage, test-time scaling is becoming an increasingly important paradigm for improving the reasoning performance of LRMs [13, 42, 54]. Prior studies have shown that LLMs' reasoning performance can continue to improve as more compute is allocated at inference time [22]. Inspired by prior work [33], we propose a simple yet effective strategy called Thinking Token based Test-time Scaling (TTTS).

**Method.** Given the set of thinking tokens identified in Section 3.1, we filter out tokens with little semantic content (*e.g.,* punctuations and single characters, see Appendix B for more details) and retain tokens like "So," "Hmm," which often indicate reflection, transition, or further thinking. Then during inference, we append one of these thinking tokens to the end of the model's initial output and allow it to continue generating additional reasoning steps.

**Experimental setup.** We evaluate TTTS using LLaMA-8B on GSM8K, MATH500, and AIME24. Specifically, we consider a controlled test-time scaling setting: given a LRM with an initial token budget, we gradually increase the token generation budget and compare the model's reasoning performance with and without TTTS.

**Results.** As shown in Figure 7, **under the same token budget, TTTS consistently outperforms the original LRM on both GSM8K and MATH500.** Notably, on GSM8K, the original LRM's

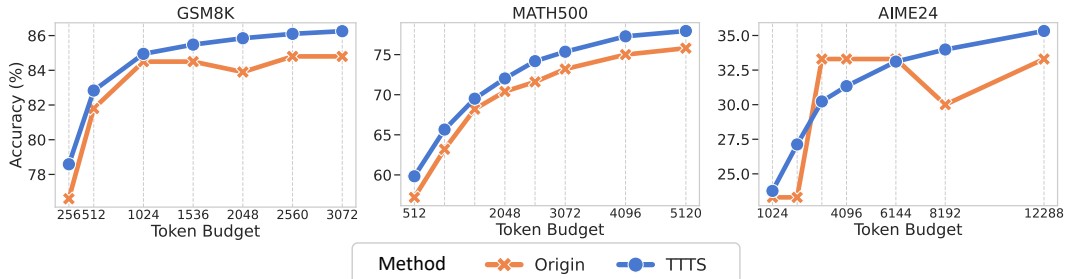

Figure 7: Reasoning performance of TTTS and the original LRMs across multiple math benchmarks under varying token budgets.

performance plateaus once the token budget exceeds 1024, whereas **TTTS continues to yield performance improvements as the token budget increases**. On the harder AIME24 benchmark, we observe that the original model's performance saturates once the token budget reaches around 3000. In contrast, although TTTS underperforms slightly at some intermediate token budgets, its performance continues to improve steadily and eventually surpasses the original model once the budget exceeds 6144 tokens. These results suggest that as more inference-time resources become available, TTTS could effectively prompt LRMs to further think, and stably improve the model's reasoning performance.

## 5 Related work

**Reasoning in LLMs.** LLMs have achieved significant advancements in understanding, particularly for complex reasoning tasks [53, 28, 45, 61]. The development of multi-step reasoning frameworks began with the chain-of-thought (CoT) paradigm [53], which introduces structured prompting to formalize explicit intermediate reasoning steps. Surprisingly, this principle is further simplified by [26], where the authors demonstrate that minimalist prompts (e.g., "Let us think step by step") could achieve comparable performance. Authors in [62] systematize problem decomposition via least-to-most prompting hierarchies. This trajectory culminated in [60] formalizing reasoning as tree-structured search processes, enabling backtracking and strategic exploration through explicit state-space modeling. Refinement Strategies also address practical limitations. Wang et al. [51] introduced self-consistency voting to mitigate output instability.

**Information Theory in LLMs.** Information theory [11] provides valuable theoretical basis for analyzing the behavior of language models [23, 12, 34], with applications spanning numerous fields: reasoning process diagnostics through quantification of unsupervised information gain [47], model optimization via information bottleneck distillation [8], systematic behavior analysis capturing dependency laws [9] and error propagation dynamics [13]. Recent extensions formalize synthetic data generation through reverse-bottleneck metrics [14], demonstrating information theory's versatility in bridging theoretical insights with engineering practices. Ren and Liu [40] show that Transformers exhibit an inductive bias toward lower-entropy representations when approximating target distributions.

**Critical Tokens in LLMs.** Prior work has shown that a small set of "critical tokens" can disproportionately affect an LLM's behavior, prompting methods to identify them [31], quantify their influence [16, 2], and mitigate their impact via selective training or pruning [30, 44]. Recent advances in LLM safety alignment have increasingly focused on the pivotal role of potential critical tokens. Zou et al. [65] propose a method to craft universal adversarial suffixes that induce aligned LLMs to generate inappropriate content. Lin et al. [29] find that after alignment, tokens like "sorry," "however," and "apolog" are learned by the model to prevent generating harmful outputs. Qi et al. [36] show that simply forcing an unaligned LLM to begin its responses with certain safe tokens can significantly improve the model's safety.

## 6 Conclusion

In this work, we systematically investigate the reasoning mechanisms of LRMs through an information-theoretic perspective. By tracking the MI evolution between intermediate represen-

tations and the golden answer, we unveil an interesting *MI peaks* phenomenon. Further, we find that these MI peaks predominantly correspond to *thinking tokens* (e.g., "Hmm," "Wait," "Therefore") that express self-reflection, logical transitions, or self-correction. Theoretically, we show that higher cumulative MI correlates with tighter bounds on model error, offering insights to the MI peaks phenomenon. Building on these analyzes, we introduce two simple, training-free methods—Representation Recycling (RR) and Thinking Token based Test-time Scaling (TTTS)—that effectively improve LRMs' reasoning performance. We hope our analyze could shed new light on the internal structure of LRM reasoning and open up new directions for inference-time reasoning enhancement.

## Acknowledgments

This research was supported by Shanghai Artificial Intelligence Laboratory, National Key Research and Development Program of China (NO. 2024YFE0203200), National Natural Science Foundation of China (No.62476277), CCF-ALIMAMA TECH Kangaroo Fund(No.CCF-ALIMAMA OF 2024008), and Huawei-Renmin University joint program on Information Retrieval. We also acknowledge the support provided by the fund for building worldclass universities (disciplines) of Renmin University of China and by the funds from Beijing Key Laboratory of Big Data Management and Analysis Methods, Gaoling School of Artificial Intelligence, Renmin University of China, from Engineering Research Center of Next-Generation Intelligent Search and Recommendation, Ministry of Education, from Intelligent Social Governance Interdisciplinary Platform, Major Innovation & Planning Interdisciplinary Platform for the "DoubleFirst Class" Initiative, Renmin University of China, from Public Policy and Decision-making Research Lab of Renmin University of China, and from Public Computing Cloud, Renmin University of China.

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

# Contents

# A  Proofs and Definitions

## A.1  Proof of Theorem 1

**Theorem 1.** *Consider a sequence of representations $\boldsymbol{h}_1, \boldsymbol{h}_2, \ldots, \boldsymbol{h}_T$ during an LLM's reasoning process, where $T$ denotes the number of total reasoning steps. Let $y$, $\hat{y}$ denote the golden answer and the LLM's prediction answer, respectively. Define $p_e = \Pr(\hat{y} \neq y)$ as the LLM's prediction error probability. Then the following inequality holds:*

$$p_e \geqslant \frac{1}{\log(|\mathcal{Y}| - 1)} \Big[ H(y) - \sum_{j=1}^{T} I\big(y; \boldsymbol{h}_j \mid \boldsymbol{h}_{<j}\big) - H_b(p_e) \Big], \tag{1}$$

*where $|\mathcal{Y}|$ is the size of the support of $y$, and $H_b(p_e)$ denote the binary entropy of $p_e$ that defined by*

$$H_b(p_e) = -p_e \log p_e - (1 - p_e) \log(1 - p_e). \tag{2}$$

*Proof.* We first define an indicator random variable $E = \mathbf{1}\{\hat{y} \neq y\}$, where $E = 1$ if $\hat{y} \neq y$, and $E = 0$ otherwise.

By the chain rule of entropy, we have:

$$\begin{aligned} H(y \mid \hat{y}) =& H(E \mid \hat{y}) + H(y \mid \hat{y}, E) \\ =& H(E \mid \hat{y}) + H(y \mid \hat{y}, E = 0) \Pr(E = 0) + H(y \mid \hat{y}, E = 1) \Pr(E = 1). \end{aligned} \tag{3}$$

Since $E = 0$ indicates $\hat{y} = y$, we have $H(y \mid \hat{y}, E = 0) = 0$. And for $H(E \mid \hat{y})$, we have:

$$H(E \mid \hat{y}) \leqslant H(E) := H_b(p_e). \tag{4}$$

Thus, we can derive:

$$H(y \mid \hat{y}) \leqslant H_b(p_e) + p_e H(y \mid \hat{y}, E = 1). \tag{5}$$

Since $E = 1$ indicates $\hat{y} \neq y$, the random variable $y$ can take at most $|\mathcal{Y}| - 1$ values given $\hat{y}$ as condition. Hence, we have [13]:

$$H(y \mid \hat{y}) \leqslant H_b(p_e) + p_e \log(|\mathcal{Y}| - 1). \tag{6}$$

Based on the definition of mutual information, we have:

$$I(y; \hat{y}) = H(y) - H(y \mid \hat{y}). \tag{7}$$

Combining Eq. (6) and Eq. (7) derives:

$$p_e \geqslant \frac{1}{\log(|\mathcal{Y}| - 1)} \Big[ H(y) - I(y; \hat{y}) - H_b(p_e) \Big]. \tag{8}$$

Consider an LLM's reasoning process, given the intermediate representations $\boldsymbol{h}_{1:T} = (\boldsymbol{h}_1, \boldsymbol{h}_2, \ldots, \boldsymbol{h}_T)$, the output $\hat{y}$ is computed as a function of these representations $\hat{y} = f(\boldsymbol{h}_{1:T})$. Thus, based on the Data Processing Inequality (DPI), we have:

$$I(y; \hat{y}) \leqslant I(y; \boldsymbol{h}_{1:T}). \tag{9}$$

Combining Eq. (8) and Eq. (9), and applying the chain rule of mutual information, we have:

$$p_e \geqslant \frac{1}{\log(|\mathcal{Y}| - 1)} \Big[ H(y) - \sum_{j=1}^{T} I\big(y; \boldsymbol{h}_j \mid \boldsymbol{h}_{<j}\big) - H_b(p_e) \Big], \tag{10}$$

which completes the proof. $\qquad\square$

## A.2 Proof of Theorem 2

**Theorem 2.** *Following the notations in Theorem 1, the following inequality holds:*

$$p_e \leqslant \frac{1}{2}\Big[H(y) - \sum_{j=1}^{T} I\big(y; \boldsymbol{h}_j \mid \boldsymbol{h}_{<j}\big)\Big]. \tag{11}$$

*Proof.* The output of a reasoning model $\hat{y}$ can be formulated as a multi-class classification task with predicted probabilities $p_i = \Pr(\hat{y} = i \mid \boldsymbol{h}_{1:T})$. According to Bayesian decision theory[5] [63], the conditional error probability is given by:

$$p_e = 1 - \max_i\{\Pr(y = i \mid \boldsymbol{h}_{1:T})\}. \tag{12}$$

For binary classification ($|\mathcal{Y}| = 2$), we have:

$$min\{p, 1 - p\} \leqslant \frac{1}{2}[-p\log p - (1 - p)\log(1 - p)]. \tag{13}$$

Then take an expectation over $p$:

$$p_e = \mathbb{E}_p[min\{p, 1 - p\}] \leqslant \frac{1}{2}\mathbb{E}_p[-p\log p - (1 - p)\log(1 - p)]. \tag{14}$$

So we derive:

$$p_e \leqslant \frac{1}{2}\mathbb{E}_{h_{1:T}}[H(y \mid \boldsymbol{h}_{1:T})] = \frac{1}{2}H(y \mid \boldsymbol{h}_{1:T}). \tag{15}$$

This extends to multiclass problems through a recursive application (see Eq. (16)).

We prove the following inequality by mathematical induction that for any $m$-class discrete probability distribution $\{p_1, \ldots, p_m\}$:

$$p_e = 1 - \max_i\{p_i\} \leqslant \frac{1}{2}H(p_1, \ldots, p_m). \tag{16}$$

*Base case* ($m = 2$): Direct verification using binary entropy function Eq. (13).

*Inductive step*: Assume validity for $m$ classes. For $m + 1$ classes, assume without loss of generality $p_{m+1} = \max_i\{p_i\}$. Consider the merged distribution $\{p_1, \ldots, p_{m-1}, p_m + p_{m+1}\}$ and apply:

1. The induction hypothesis:

$$1 - (p_m + p_{m+1}) \leqslant \frac{1}{2}H(p_1, \ldots, p_{m-1}, p_m + p_{m+1}). \tag{17}$$

2. The grouping axiom [4]:

$$H(p_1, \ldots, p_{m+1}) = H(p_1, \ldots, p_m + p_{m+1}) + (p_m + p_{m+1})H\left(\frac{p_m}{p_m + p_{m+1}}, \frac{p_{m+1}}{p_m + p_{m+1}}\right). \tag{18}$$

3. Binary entropy bound for the final term:

$$1 - \frac{p_{m+1}}{p_m + p_{m+1}} \leqslant \frac{1}{2}H\left(\frac{p_m}{p_m + p_{m+1}}, \frac{p_{m+1}}{p_m + p_{m+1}}\right). \tag{19}$$

Combining Eq. (17), Eq. (18) and Eq. (19) completes the induction:

$$\frac{1}{2}H(p_1, \ldots, p_{m+1}) = \frac{1}{2}H(p_1, \ldots, p_m + p_{m+1}) + \frac{1}{2}(p_m + p_{m+1})H\left(\frac{p_m}{p_m + p_{m+1}}, \frac{p_{m+1}}{p_m + p_{m+1}}\right)$$

$$\geqslant 1 - (p_m + p_{m+1}) + (p_m + p_{m+1})(1 - \frac{p_{m+1}}{p_m + p_{m+1}})$$

$$= 1 - p_{m+1}$$

$$= 1 - \max_i\{p_i\}.$$

Thus, we have proved the Eq. (16).

Taking expectation over $h_{1:T}$ in Eq. (12) and applying the Eq. (16), we have

$$p_e = \mathbb{E}_{h_{1:T}}[1 - \max_i\{\Pr(y = i|h_{1:T})\}].$$

$$\leqslant \frac{1}{2}\mathbb{E}_{h_{1:T}}[H(y|h_{1:T})]$$

$$= \frac{1}{2}H(y|h_{1:T})$$

$$= \frac{1}{2}\left[H(y) - \sum_{j=1}^{T} I(y; h_j \mid h_{<j})\right],$$

which completes the proof. $\qquad\square$

### A.3 Definitions

**Definition 3** (Mutual Information [4, 27]). *Given two continuous random variables X and Y , the mutual information is defined as:*

$$I(X;Y) = \int_Y \int_X p(x,y) \log \frac{p(x,y)}{p(x)p(y)} dx dy, \tag{20}$$

*where $p(x,y)$ denotes the joint probability density function of $X$ and $Y$; $p(x)$, $p(y)$ denotes the marginal probability density functions of $X$ and $Y$, respectively.*

**Definition 4** (Hilbert-Schmidt Independence Criterion (HSIC) [18]). HSIC *is the Hilbert-Schmidt norm of the cross-covariance operator between the distributions in Reproducing Kernel Hilbert Space (RKHS). Formally:*

$$\mathrm{HSIC}(X,Y) = \mathbb{E}_{XYX'Y'}\left[k_X\left(X,X'\right)k_Y\left(Y,Y'\right)\right] + \mathbb{E}_{XX'}\left[k_X\left(X,X'\right)\right]\mathbb{E}_{YY'}\left[k_Y\left(Y,Y'\right)\right]$$
$$-2\mathbb{E}_{XY}\left[\mathbb{E}_{X'}\left[k_X\left(X,X'\right)\right]\mathbb{E}_{Y'}\left[k_Y\left(Y,Y'\right)\right]\right], \tag{21}$$

*where $X'$, $Y'$ are independent copies of $X$, $Y$, respectively, and $k_X$, $k_Y$ are kernel functions.*

## B Experimental Implementation Details

**Practical implementation of HSIC.** Due to the difficulty of accurately computing MI in high-dimensional spaces [27, 35, 13], we employ the HSIC to estimate MI. Following [32, 38, 13], the empirical HSIC from Definition 4 is computed as

$$\mathrm{HSIC}(X,Y) = \frac{1}{(n-1)^2}\,\mathrm{tr}\big(K_X\,H\,K_Y\,H\big), \tag{22}$$

where $K_X$ and $K_Y$ are kernel matrices with entries

$$K_{X_{ij}} = k_X(x_i, x_j), \quad K_{Y_{ij}} = k_Y(y_i, y_j),$$

and $H = I - \frac{1}{n}\mathbf{1}\mathbf{1}^\top$ is the centering matrix. Consistent with [32, 38, 13], we adopt the Gaussian kernel to implement the kernel:

$$k(\mathbf{x}, \mathbf{y}) = \exp\left(-\frac{\|\mathbf{x} - \mathbf{y}\|^2}{2\sigma^2}\right), \tag{23}$$

where the bandwidth $\sigma$ is selected by grid search over the range $[50, 400]$.

**Datasets.** *1) Evaluation of LRMs' reasoning performance.* We select three widely-used math reasoning benchmarks to evaluate the reasoning capabilities of LRMs, ordering from easy to hard: GSM8K [10], MATH500 [28], and AIME24 [1]. We adopt the evaluation framework provided by Qwen2.5-Math [58]. To ensure the reproducibility of our results, we fix the temperature to 0 in all experiments. *2) Observing the MI trajectories during LRMs' reasoning process.* We use the training set of the MATH dataset [20]. Specifically, we randomly sample 100 instances to compute MI along the reasoning trajectories.

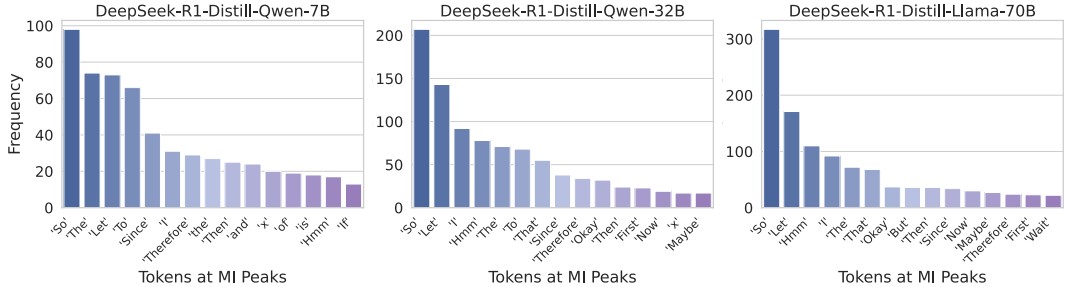

Figure 8: Frequency distribution of tokens at MI peaks for DeepSeek-R1-Distill-Qwen-7B, DeepSeek-R1-Distill-Qwen-32B, and DeepSeek-R1-Distill-Llama-70B.

**Models.** We conduct experiments on DeepSeek's R1 series models [19] and QwQ-32B [46]. For DeepSeek's R1 series models, we pair each LRM with its corresponding non-reasoning LLM counterpart as follows: DeepSeek-R1-Distill-Qwen-7B and Qwen2.5-Math-7B [58], DeepSeek-R1-Distill-Llama-8B and Llama-3.1-8B [17], DeepSeek-R1-Distill-Qwen-14B and Qwen2.5-14B [57], DeepSeek-R1-Distill-Qwen-32B and Qwen2.5-32B [57], DeepSeek-R1-Distill-Llama-70B and Llama-3.3-70B-Instruct [17]. As observed, all LRMs in the R1 series are trained from foundation LLMs, except for DeepSeek-R1-Distill-Qwen-7B, which is trained from a math-specialized LLM. As for QwQ-32B, existing public report [46] has not disclosed which specific LLM it was trained from. All experiments are conducted on four NVIDIA A100 GPUs.

**More implementation details.** For all experiments involving MI computation, we extract the representation from the *last layer* of the model. We concentrate on the *last layer* since higher layers have been shown to encode more semantic content [64, 41] and the *last layer* directly influence the model's output text [37]. For TTTS in Section 4.2, to ensure that the model begins continuation with semantically meaningful tokens, we filter out tokens with little semantic information, such as punctuation, single characters, etc. In this way, the resulting token list is: [So, Let, Hmm, I, Okay, First, Wait, But, Now, Then, Since, Therefore, If, Maybe, To]. All experiments are conducted on four NVIDIA A100 GPUs.

## C Discussions

**Limitations.** This work has several limitations. First, we analyze the MI dynamics of LRMs at the token level. Alternative granularities such as dividing reasoning steps by semantic units or logical steps may reveal additional insights. Second, while we observe the interesting MI peaks phenomenon and provide insights into the reasoning mechanisms of LRMs, the underlying mechanisms that give rise to these peaks remain underexplored. We leave a deeper analysis of their origin to future work. We hope that our work will inspire further research along these directions and contribute to a deeper understanding of the reasoning process in LRMs.

**Broader impacts.** This work contributes to a deeper understanding of the reasoning mechanisms in LRMs. We first observe the MI peaks phenomenon during LRMs' reasoning process, and then propose two simple training-free methods to enhance LRMs' reasoning performance based on the findings. These analyzes may have positive impacts by making AI systems more transparent and effective. However, there are also potential risks. If used carelessly, the same methods could be applied to manipulate outputs or reinforce biased thinking patterns. It is important to consider these concerns when applying our techniques and to encourage responsible use through further study and monitoring.

**Discussion on Tokens at MI Peaks.** As shown in Figure 4 in the main text and Figure 8 in the appendix, different LRMs exhibit slightly different token frequency patterns at MI peaks. For models trained from foundation LLMs, *i.e.,* DeepSeek-R1-Distill-Llama-8B, DeepSeek-R1-Distill-Qwen-14B, DeepSeek-R1-Distill-Qwen-32B, and DeepSeek-R1-Distill-LLaMA-70B, the frequently occurring tokens include So, Let, Hmm, The, and Okay. And for DeepSeek-R1-Distill-Qwen-7B, which is trained from a math-specialized LLM, tokens such as So, The, Let, To, and, and Since are more prominent. For QwQ-32B, tokens like To, the, we, and Let appear more frequently. Semantically, these tokens commonly express reasoning-related functions such as initiating thinking

Table 3: The AOM metric of MI sequences between LRMs and their corresponding non-reasoning LLMs on GPQA and MedQA.

| Model | Type | GPQA | MedQA |
|---|---|---|---|
| Llama-3.1-8B | Origin | 3.5193 | 3.6337 |
| | Reasoning | 4.0809 | 3.8436 |
| Qwen2.5-14B | Origin | 2.7671 | 2.4355 |
| | Reasoning | 2.9444 | 2.6281 |

(So, Hmm), logical transition (Since, Therefore), or discourse structuring (Let, Then, To), which likely help facilitate the model's continued reasoning. We hypothesize that the distribution of tokens at MI peaks may be influenced by factors such as the nature of the foundation LLM, the reasoning-intensive training paradigm, etc. We leave a deeper investigation of the relationship among MI-peak token distributions, foundation LLM characteristics, reasoning-intensive training paradigms, and model reasoning performance to future work.

**Further discussion on thinking token suppression (Section 3.2, Figure 5).** As shown in Figure 3.2, while the overall trend indicates that LRMs' reasoning performance degrades as more thinking tokens are suppressed, the decline is not strictly monotonic. In some cases, performance improves temporarily. We conduct an empirical analysis to better understand this phenomenon. Specifically, we observe that when certain tokens are suppressed, the model tends to adopt alternative expressions to convey similar meanings. For instance, when the generation of the token "Wait" is suppressed, the model may instead produce phrases like "But wait", which could lead to slight improvements in performance. The observed performance fluctuations across different numbers of suppression tokens further support that these thinking tokens play a critical role in LRMs' reasoning capabilities.

**Discussion on Theorem 1 and Token Length.** In the main body, Theorems 1 and 2 are intended to provide theoretical insights into the MI peaks phenomenon. Specifically, they suggest that when the token length $T$ is fixed, the presence of MI peaks may lead to a higher cumulative MI, and thus potentially a lower error probability. This implication is also partially supported by the experimental results in Figure 3. Therefore, Theorems 1 and 2 help establish a theoretical connection between the MI peaks phenomenon and reasoning performance. However, the current formulation of Theorem 1 may be misinterpreted if extended to arbitrarily large $T$, as it does not explicitly account for the natural limitations of reasoning in practice. First, there exists an upper bound on the total amount of information that can be extracted from the input:

$$\sum_{j=1}^{T} I(y; h_j \mid h_{<j}) = I(y; h_{1:T}) \leqslant \min\{H(y), H(h_{1:T})\} \leqslant C,$$

where $C$ is a constant. This bound reflects that increasing the number of reasoning steps cannot provide unlimited additional information. Second, reasoning performance in practice does not improve monotonically with longer sequences. As $T$ increases, noise may accumulate, potentially leading to performance degradation due to distraction or loss of coherence. To capture this effect, Eq. (1) can be extended with an additional term $+f(T, \sigma)$, where $\sigma$ denotes the noise introduced during step-wise reasoning. The function $f$ can flexibly model different forms of error accumulation, including super-linear [13], approximately linear [55], and more complex behaviors [3, 25]. Together, these considerations provide a more faithful description of the relationship between cumulative MI, reasoning length, and prediction error in practice.

# D   Additional Experimental Results

## D.1   Additional Experiments on Other Reasoning Domains

To further examine the generality of the MI peaks phenomenon beyond mathematical reasoning, we additionally conduct experiments on two reasoning-intensive benchmarks: GPQA [39] and MedQA [24]. We evaluate both DeepSeek-R1-Distill-Llama-8B and DeepSeek-R1-Distill-Qwen-14B, along with their corresponding base models. The results in Table 3 show that the MI peaks phenomenon consistently persists across these domains.

Moreover, we examine the tokens corresponding to MI peaks. For GPQA, the set of tokens at MI peaks largely overlaps with that obtained from the MATH dataset in the main text. For MedQA, while common tokens such as *"Let"*, *"So"*, and *"But"* remain prominent, we also observe additional tokens such as *"Admin"*, *"Perform"*, *"She"*, and *"He"*, which may reflect the specific characteristics of the medical domain.

These additional experiments further support the generality of the MI peaks phenomenon, indicating that it is not confined to mathematical problem-solving but also emerges in other domains that require complex reasoning.

## D.2 MI Peaks in LRMs

Figures 9–20 illustrate the MI trajectories of various LRMs across more data samples.

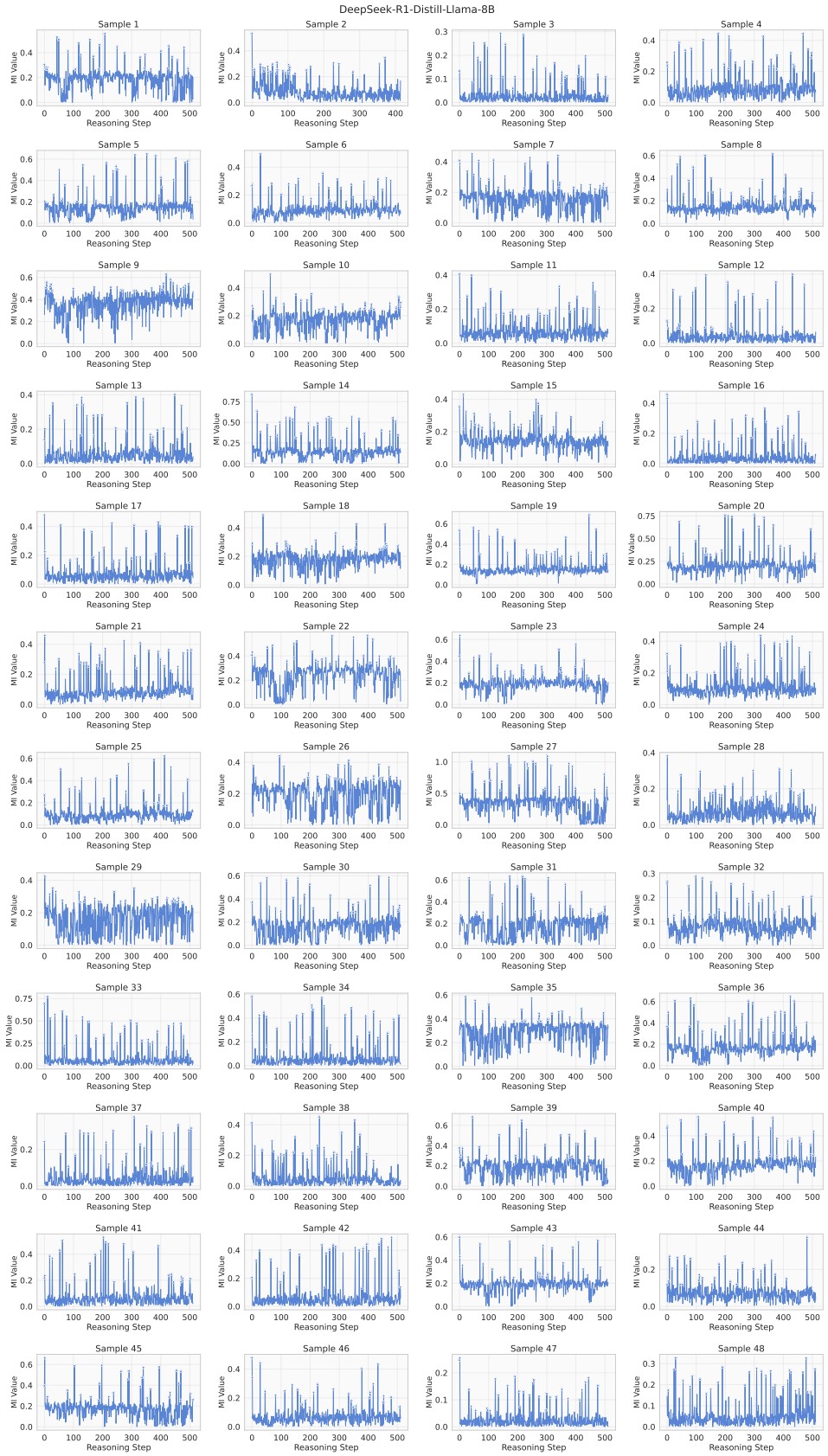

Figure 9: MI trajectories of DeepSeek-R1-Distill-Llama-8B.

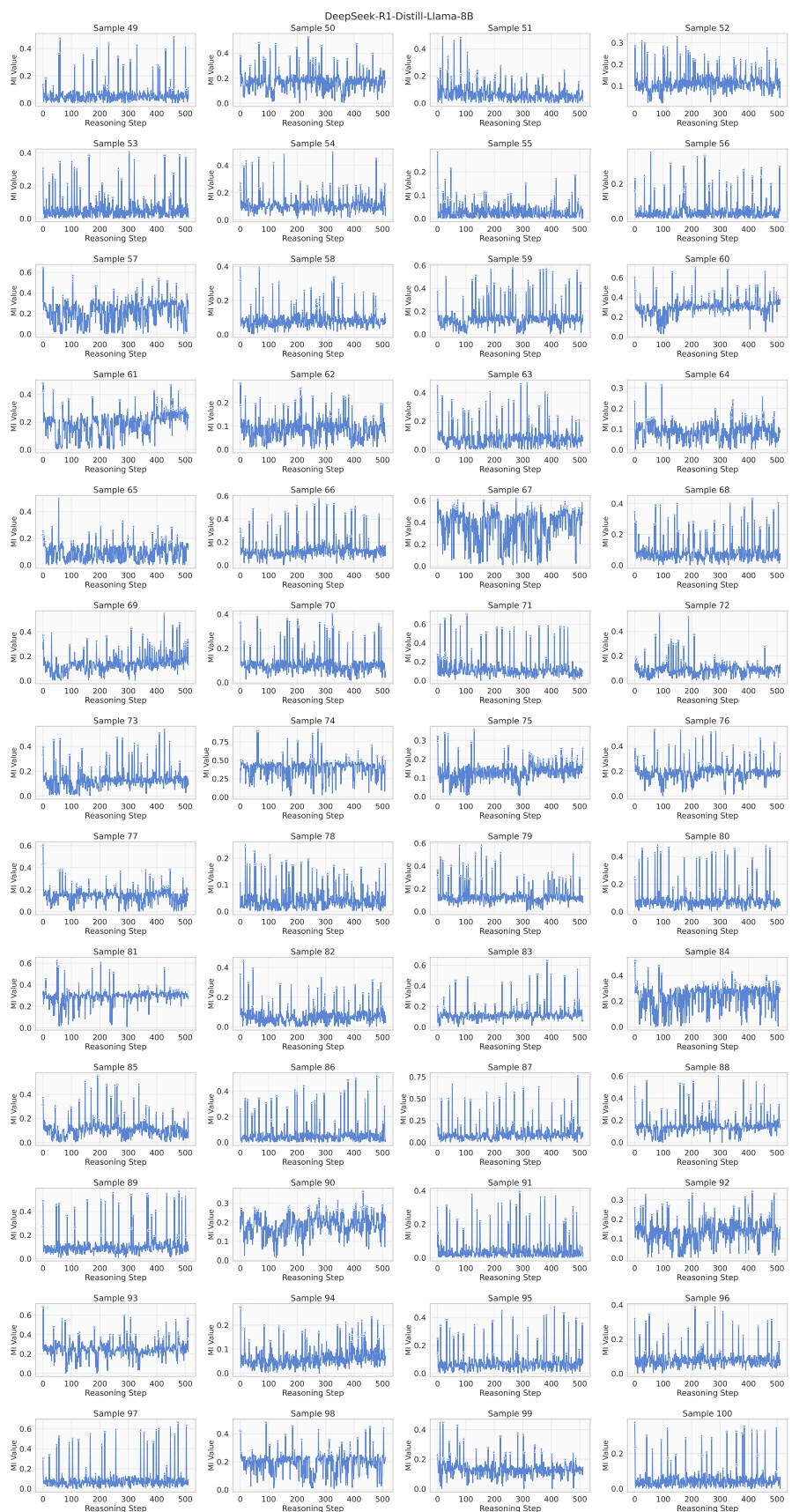

Figure 10: (Continued) MI trajectories of DeepSeek-R1-Distill-Llama-8B.

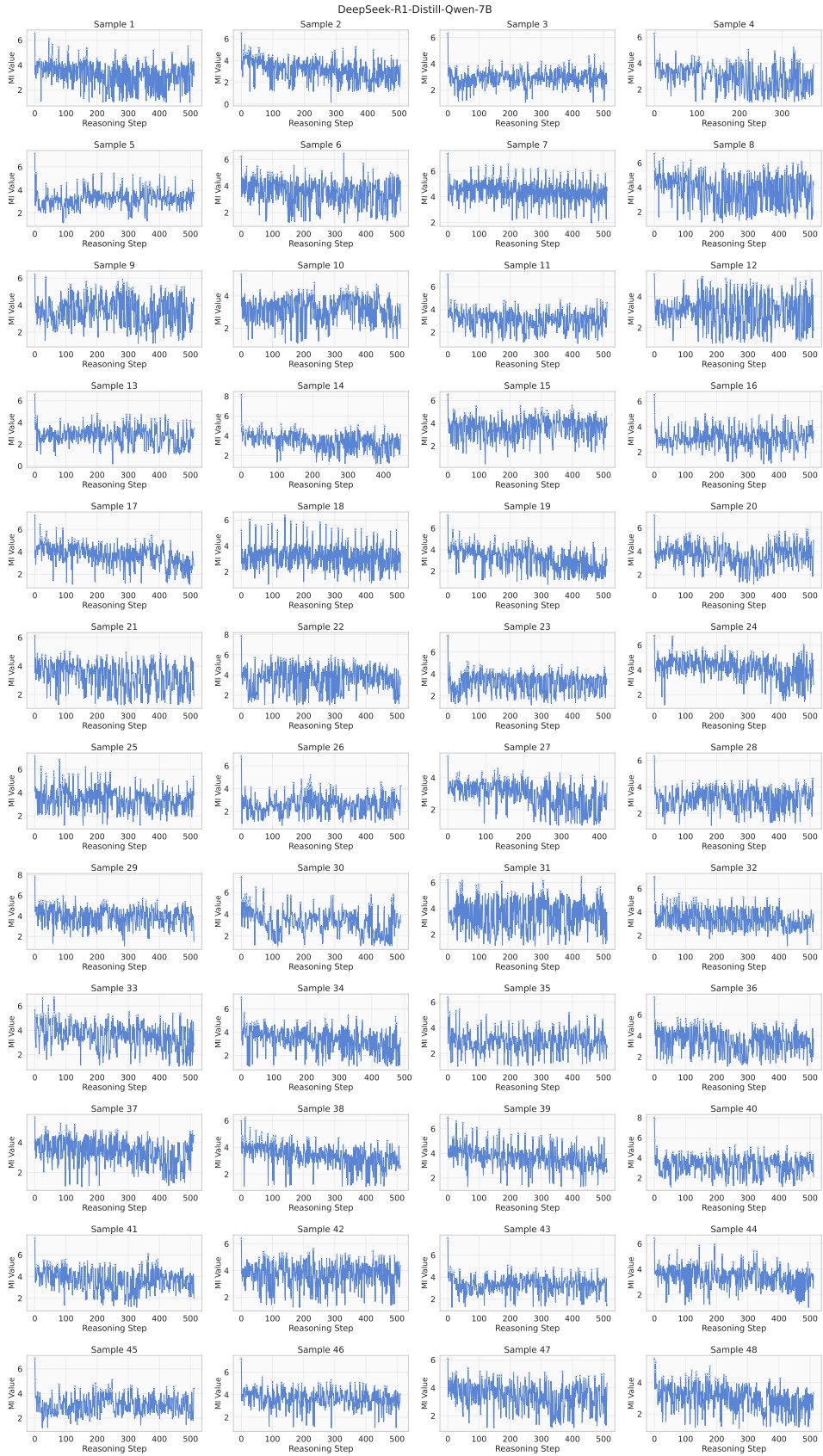

Figure 11: MI trajectories of DeepSeek-R1-Distill-Qwen-7B.

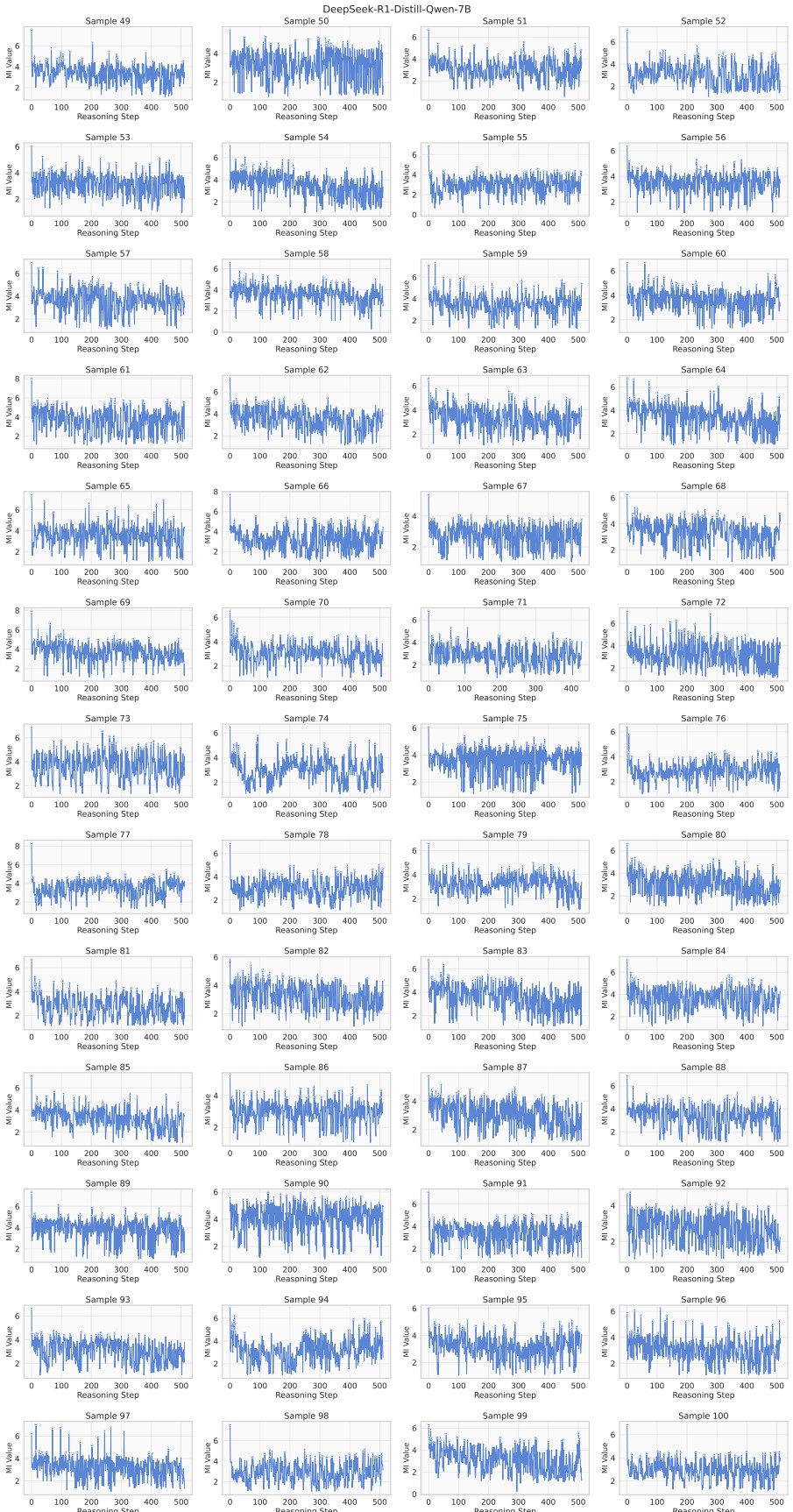

Figure 12: (Continued) MI trajectories of DeepSeek-R1-Distill-Qwen-7B.

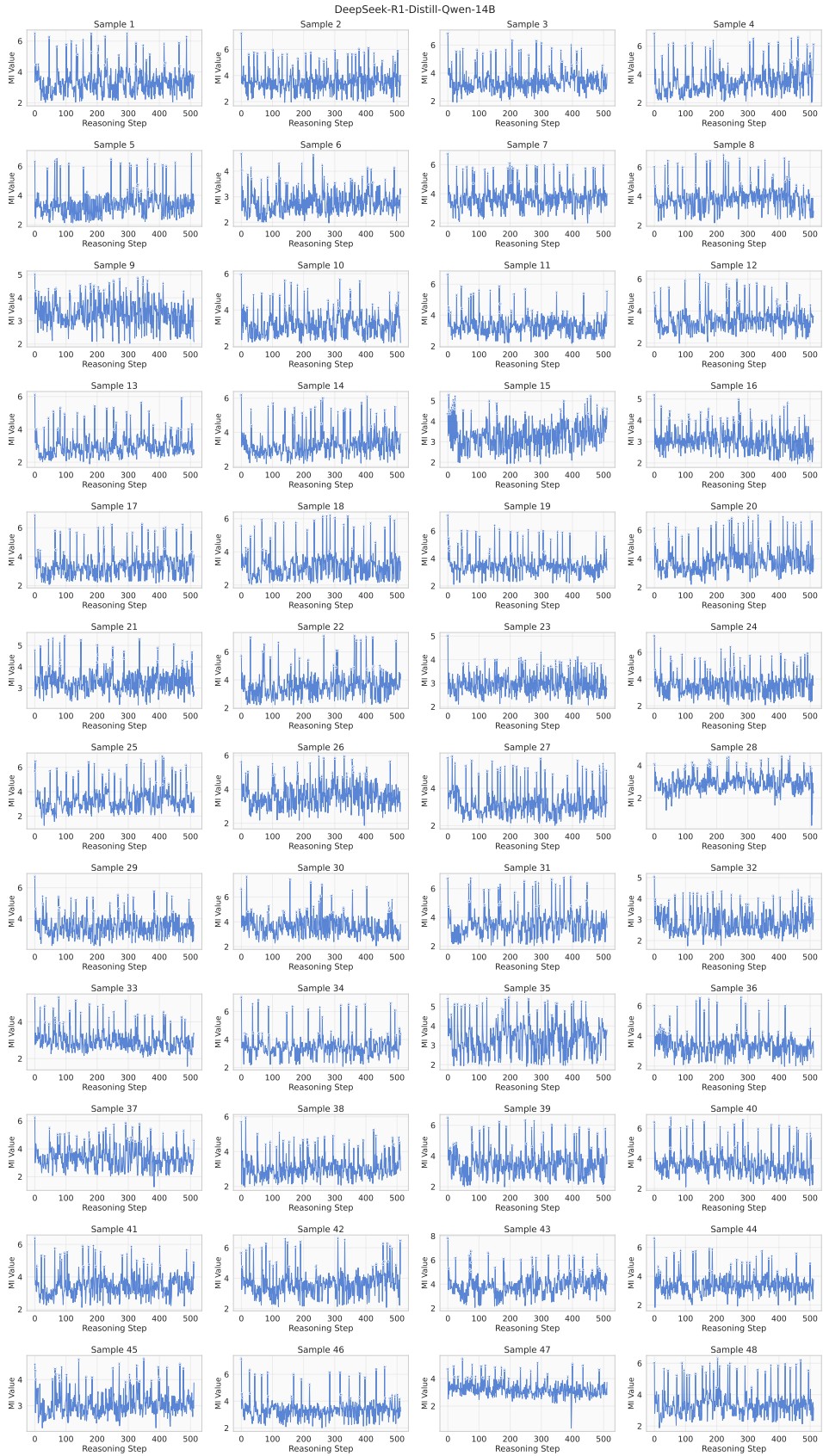

Figure 13: MI trajectories of DeepSeek-R1-Distill-Qwen-14B.

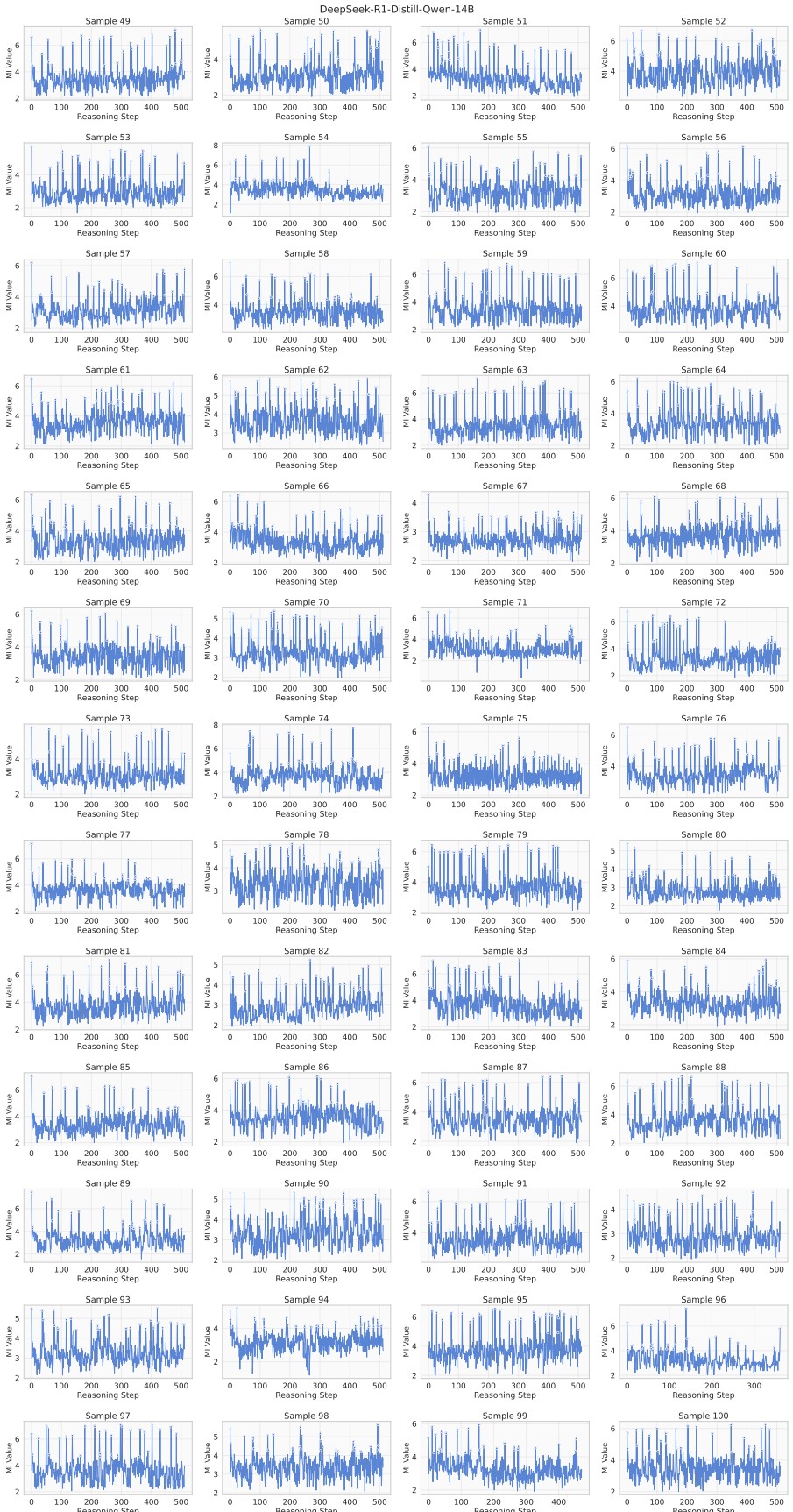

Figure 14: (Continued) MI trajectories of DeepSeek-R1-Distill-Qwen-14B.

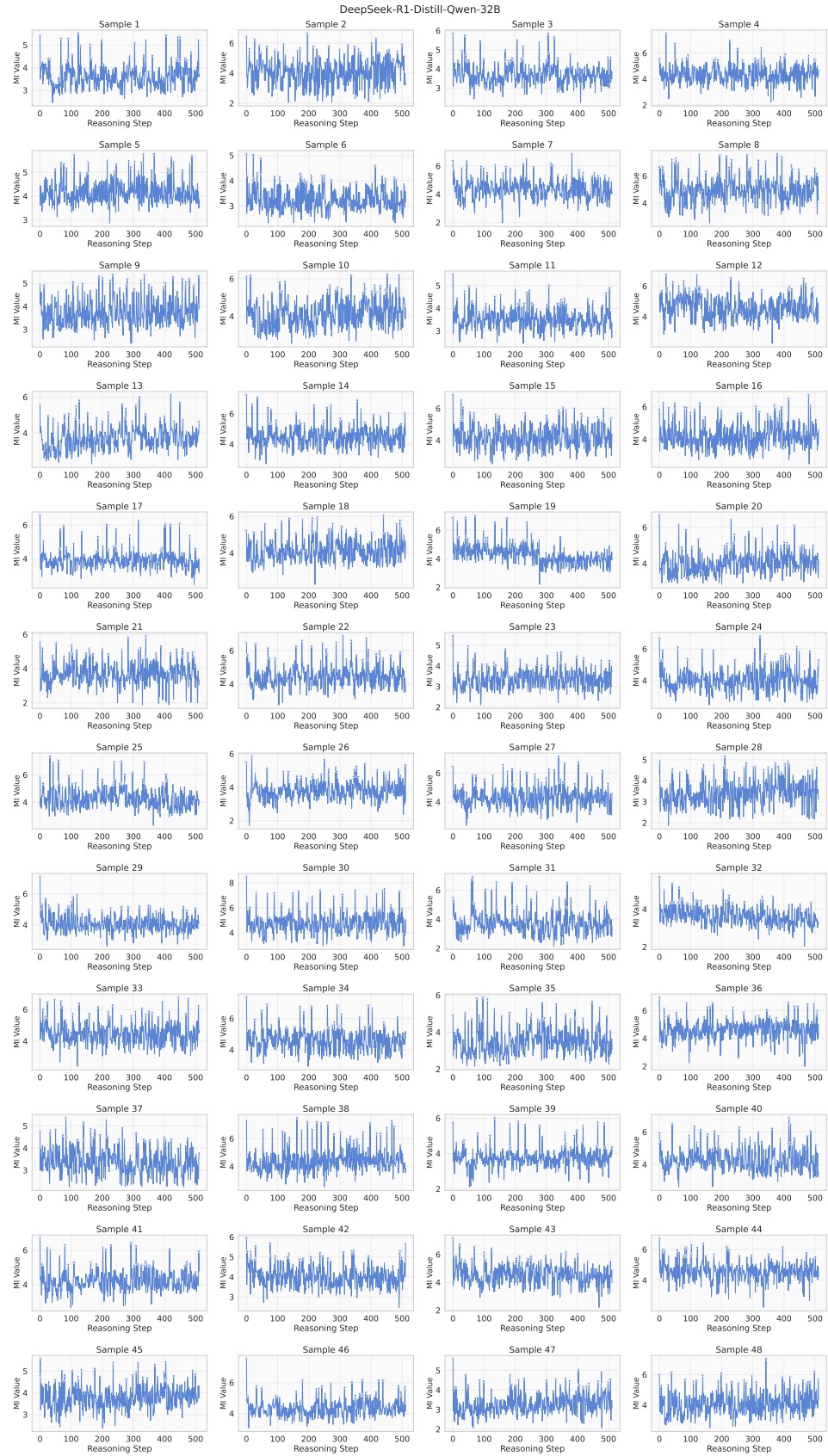

Figure 15: MI trajectories of DeepSeek-R1-Distill-Qwen-32B.

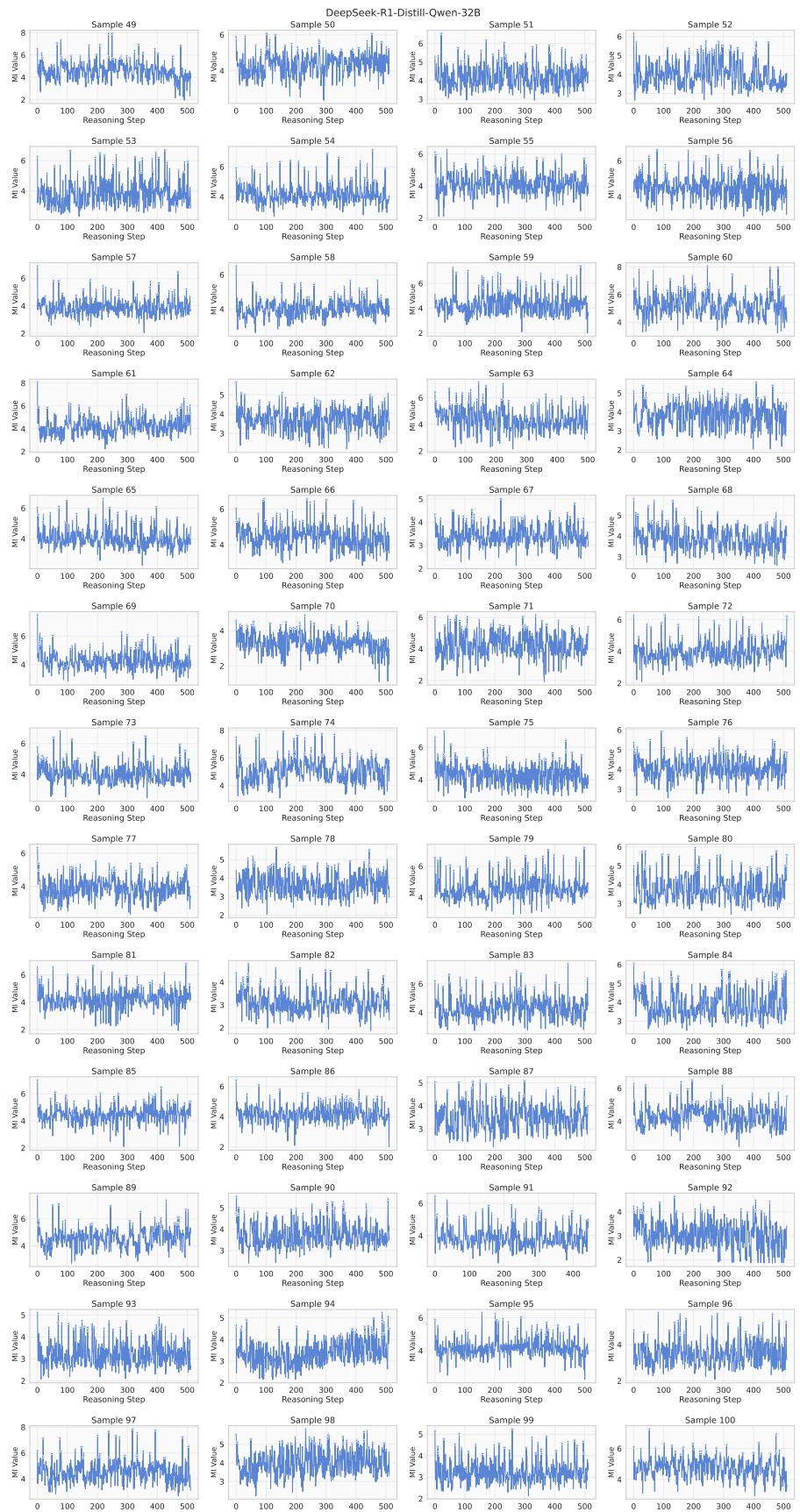

Figure 16: (Continued) MI trajectories of DeepSeek-R1-Distill-Qwen-32B.

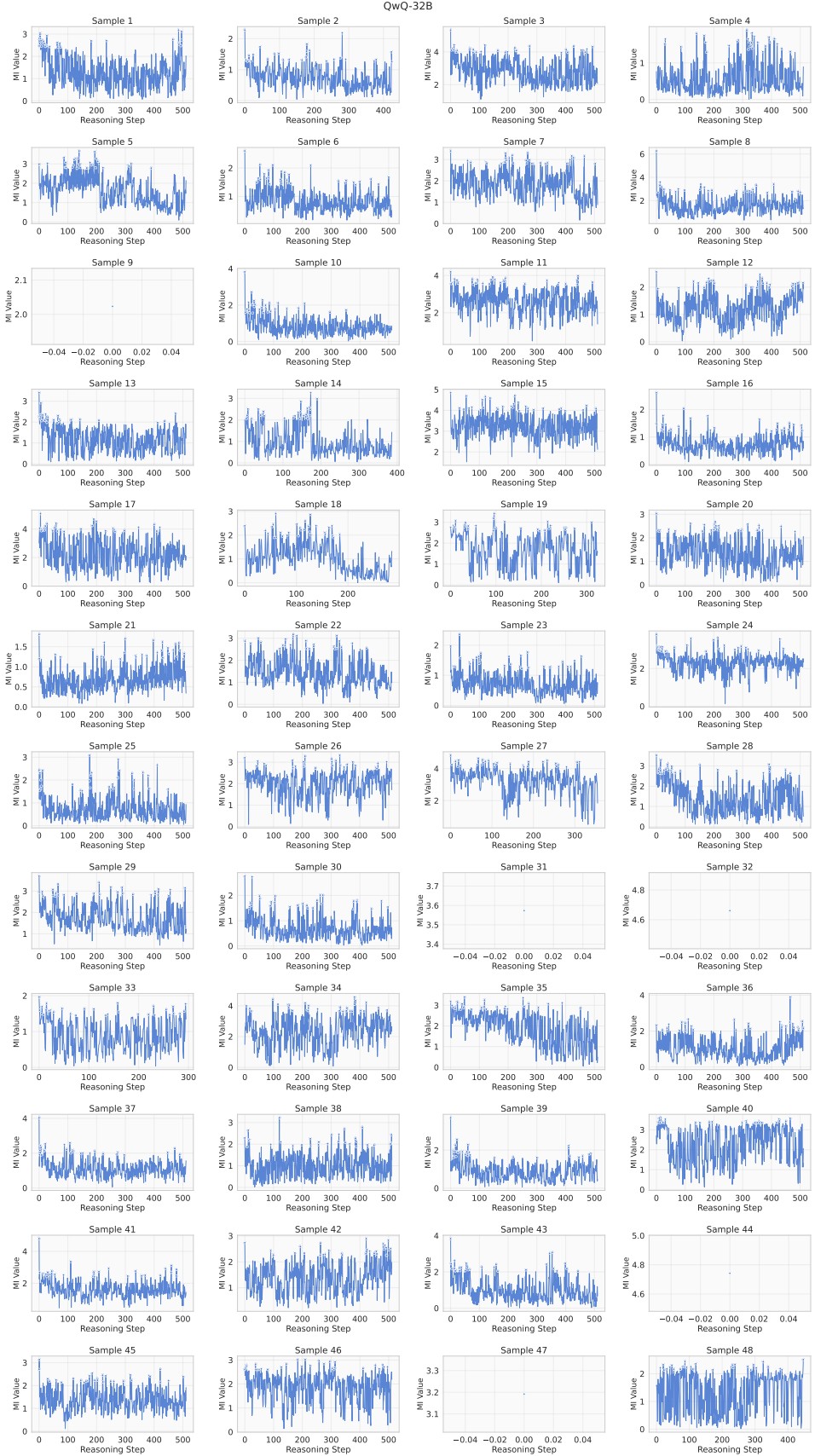

Figure 17: MI trajectories of QwQ-32B.

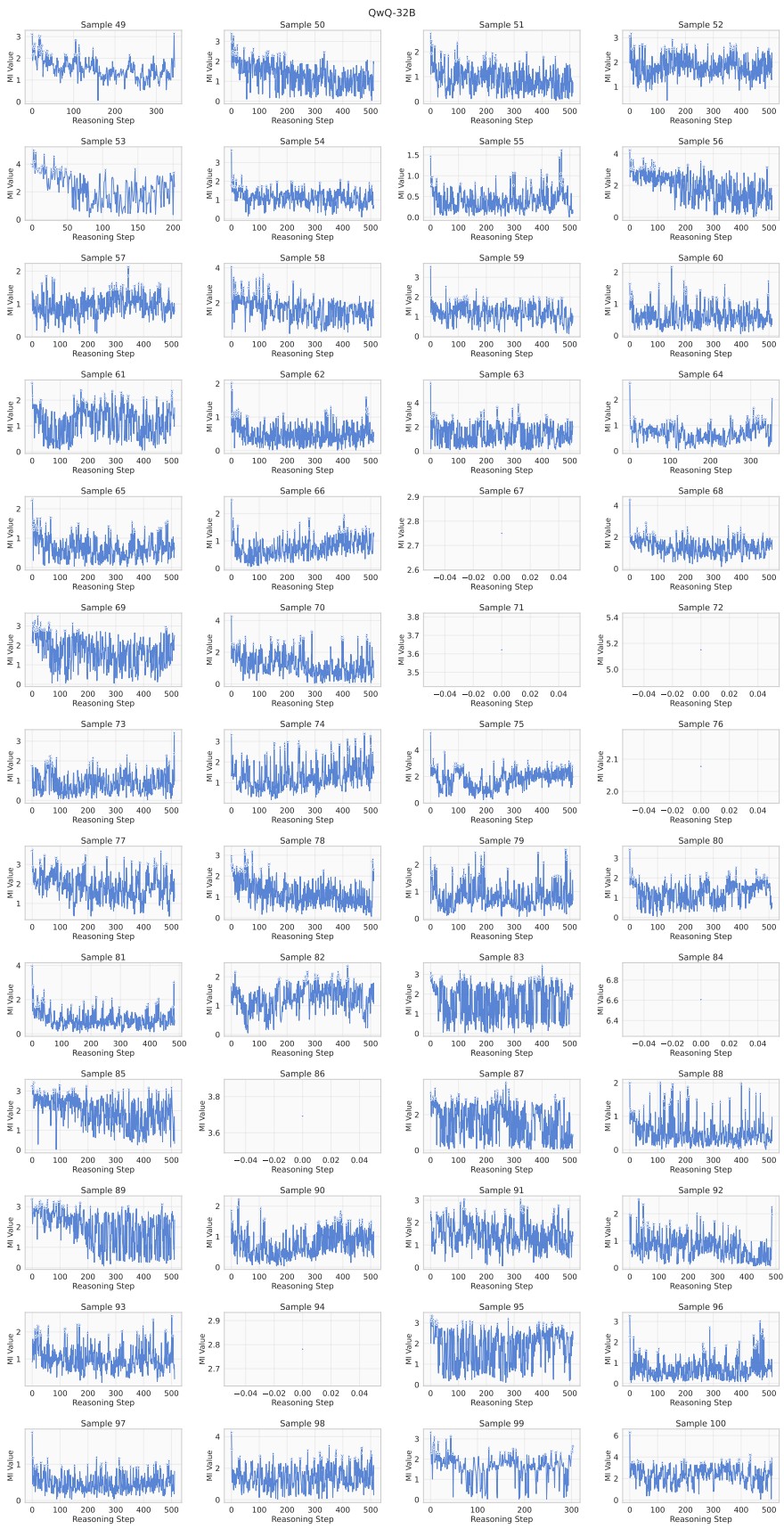

Figure 18: (Continued) MI trajectories of QwQ-32B.

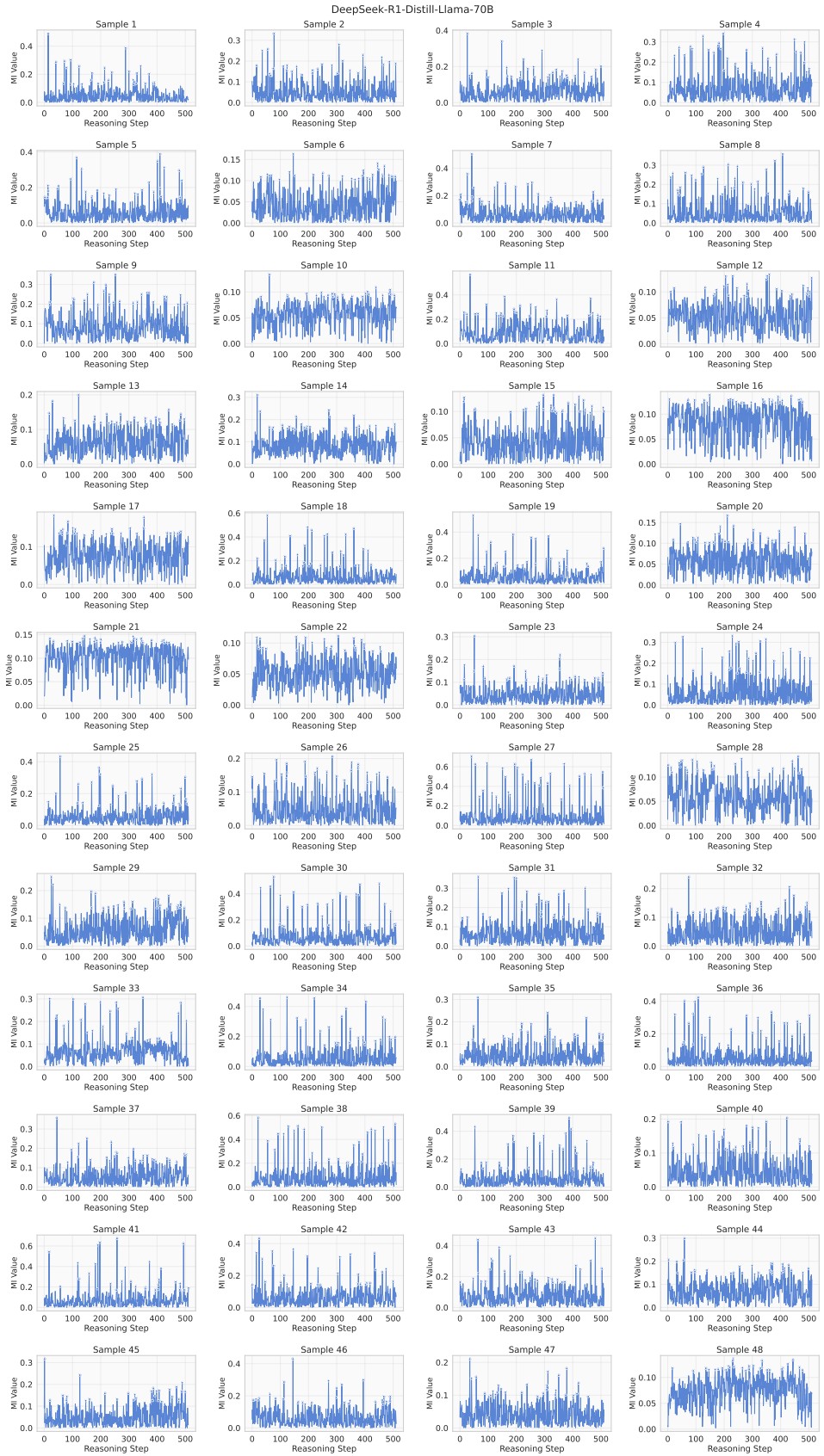

Figure 19: MI trajectories of DeepSeek-R1-Distill-Llama-70B.

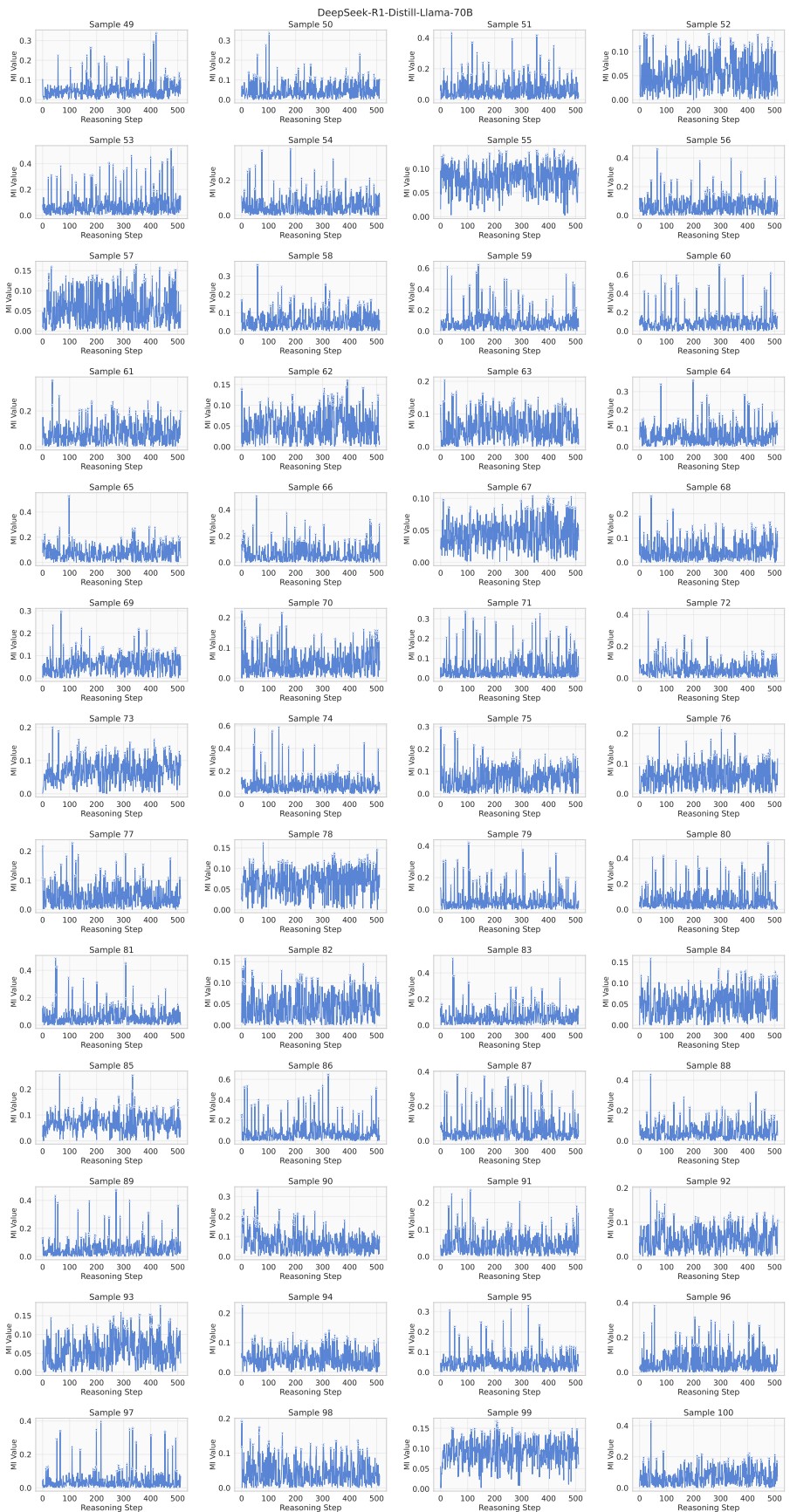

Figure 20: (Continued) MI trajectories of DeepSeek-R1-Distill-Llama-70B.

