# OpenReview forum: "Demystifying Reasoning Dynamics with Mutual Information: Thinking Tokens are Information Peaks in LLM Reasoning"
_NeurIPS.cc/2025/Conference — NeurIPS 2025 poster_

### Official Review · Reviewer_Pihp · 2025-06-12

**Clarity:** 2
**Significance:** 4
**Originality:** 3
**Rating:** 5
**Confidence:** 3

**Summary:**

The paper analyzes how the mutual information (MI) between intermediate representations and the ground truth answer evolves in reasoning models. The MI is estimated by computing the HSIC criterion between the internal representation of each step, and the LLM representation of the answer. It is observed that the MI sequence sharply peaks at specific steps throughout the reasoning process. Such peaks often correspond to thinking tokens (so,wait,hmm,therefore), aligning with the popular belief that self-correction or reflection during thinking boosts reasoning ability. Suppressing thinking tokens are demonstrated to harm reasoning performance, justifying their importance. Moreover, two methods (reusing representations and forcing thinking token generation) are proposed to improve reasoning performance at test time.

**Questions:**

See Weaknesses

**Ethical Concerns:**

["NO or VERY MINOR ethics concerns only"]

**Final Justification:**

The authors have addressed my concerns on statistical significance and whether MI behaves as expected (measuring concatenated MI). Overall, this is a strong empirical paper and raises important points which merit further study from an information theory perspective.

**Limitations:**

Yes (Appendix C)

**Quality:**

3

**Strengths And Weaknesses:**

**Strengths**

The paper enables a novel quantitative analysis to a very timely topic. The proposed MI experiments and RR/TTTS methods quantifies the crucial importance of thinking tokens, or transitions in reasoning paths. The methods not only make reasoning amenable to information theoretic analyses but also serve as a guideline to enable more effective inference time scaling.

**Weaknesses/Questions**

It is intuitively not very clear to me what exactly MI is measuring. If MI is to be taken as how informative the hidden representation at each step is about the correct answer, should we not expect MI to increase throughout the reasoning process as the model arrives closer to the answer? (Perhaps this can be quantified instead as the MI between the answer and the concatenated representations up to time $T$.) Why does MI increase sharply at thinking tokens but fall back down afterwards? The paper does a great job at showing the importance of thinking tokens but could have more discussion on what this signifies or why this happens.

It should be mentioned that Theorem 1 basically follows from an application of Fano's inequality (Eq.6). Also, the result is stated with the conditional MI $I(y;h_j|h_{<j})$, so the precise connection with the experiments is unclear. Can we also compute the conditional MI in the experiments?

Some results may not be statistically significant, in particular the gaps between the original and proposed models in Figure 7 are not very large and would benefit from adding deviations/error bars over a couple of trials.

---

> ### Author Rebuttal · Authors · 2025-07-31
>
> Thank you for your great efforts in reviewing this paper. We will try our best to answer all your questions. **Please let us know if you still have further concerns, or if you are not satisfied with the current responses, so that we can further update the response ASAP.**
>
> ---
>
> **Q1**: "It is intuitively not very clear to me what exactly MI is measuring. If MI is to be taken as how informative the hidden representation at each step is about the correct answer, should we not expect MI to increase throughout the reasoning process as the model arrives closer to the answer? (Perhaps this can be quantified instead as the MI between the answer and the concatenated representations up to time .) Why does MI increase sharply at thinking tokens but fall back down afterwards? The paper does a great job at showing the importance of thinking tokens but could have more discussion on what this signifies or why this happens."
>
> **A1**: Thank you for your insightful comment!
>
> - **Intuitively, your understanding of "what exactly MI is measuring" is completely correct**, i.e., "MI is to be taken as how informative the hidden representation at each step is about the correct answer." However, what we compute is the MI between the representation of **each generated token** and the answer. Since **the representation of a single token may not contain all information up to the current step**, the MI of the whole sequence is not necessarily monotonically increasing.
>
> - Following your suggestion, we **conduct new experiments** to quantify the MI between the answer and the **concatenated representations up to each time step**. Specifically, we follow the same experimental setting as in the paper and use the DeepSeek-R1-Distill-Qwen-7B model on the training split of the MATH dataset. We compute Spearman’s ρ and Kendall’s τ to measure the monotonicity of the MI sequence. Both metrics take values in [−1,1], with **values closer to +1 indicating stronger monotonic increasing trends**.
>
>    | **Spearman’s ρ** |         | **Kendall’s τ** |          |
>    | ---------------- | ------- | --------------- | -------- |
>    | value            | p-value | value           | p-value  |
>    | 0.9204           | 0.0051  | 0.8150          | $4.2e^{-11}$ |
>
>    The results show that when computing MI between the concatenated representations and the answer, **the MI sequence indeed increases as reasoning progresses, consistent with your insight**.
>
> - Regarding "*Why does MI increase sharply at thinking tokens but fall back down afterwards"*, we propose a possible explanation from the perspective of the definition of MI: $I(X;Y) = H(X) - H(X|Y)$, where $H(X)$ measures the uncertainty of $X$. During reasoning, when some tokens represent transitions or explore new reasoning paths (e.g., *“Wait”*, *“But”*), they may have higher uncertainty. In contrast, most other tokens simply continue along an already established reasoning path in a more deterministic manner (e.g., *“1+1=2”*, *“and”*).
>
> - Finally, thank you for your recognition of our paper. We fully agree with your point that it is interesting and important to further investigate why the MI peaking phenomenon occurs (which we also discussed in the limitations section). We hope our work can inspire future explorations in this direction.
>
> ---
>
> **Q2**: "It should be mentioned that Theorem 1 basically follows from an application of Fano's inequality (Eq.6). Also, the result is stated with the conditional MI , so the precise connection with the experiments is unclear. Can we also compute the conditional MI in the experiments?"
>
> **A2**: Thank you for your comment.
>
> - We will make the connection to Fano’s inequality explicit in the revised manuscript.
> - Regarding the conditional MI $I(y; h_j | h_{<j})$ in Theorem 1, we follow your suggestion to improve clarity. By the chain rule and the definition of mutual information, we can rewrite: $I(y; h_j | h_{<j}) = I(y; h_j) + I(h_j; h_{<j} | y) -  I(h_j; h_{<j})$. Since in the setting of this paper, we assume that y is independent  to $h_j$, $h_{<j}$, respectively (only the input X affects $h_j$, $h_{<j}$. Therefore, $I(h_j; h_{<j} | y) = I(h_j; h_{<j})$. As a result, the conditional MI in Theorem 1 $I(y; h_j | h_{<j})$ can be directly replaced by $I(y; h_j)$。
>
> ---
>
> **Q3**: "Some results may not be statistically significant, in particular the gaps between the original and proposed models in Figure 7 are not very large and would benefit from adding deviations/error bars over a couple of trials."
>
> **A3**: Thank you for your suggestion. **We have followed your instructions to conduct new experiments**, running both the origin and TTTS settings in Figure 7 **three times and reporting the mean and standard deviation**. The results are as follows:
>
> MATH500:
>
> | **Budget** | **Origin**    | **TTTS**          |
> | ---------- | ------------- | ----------------- |
> | 512        | 0.5200±0.0214 | **0.5944±0.0076** |
> | 1024       | 0.5827±0.0155 | **0.6404±0.0083** |
> | 1536       | 0.6273±0.0057 | **0.6788±0.0071** |
> | 2048       | 0.6620±0.0085 | **0.7049±0.0071** |
> | 3072       | 0.6907±0.0203 | **0.7347±0.0072** |
> | 4096       | 0.7207±0.0077 | **0.7452±0.0115** |
> | 5120       | 0.7433±0.0098 | **0.7649±0.0069** |
> | 7168       | 0.7633±0.0125 | **0.7792±0.0094** |
>
> GSM8K：
>
> | **Budget** | **Origin**    | **TTTS**          |
> | ---------- | ------------- | ----------------- |
> | 256        | 0.6967±0.0054 | **0.7292±0.0076** |
> | 512        | 0.7627±0.0079 | **0.7837±0.0072** |
> | 1024       | 0.7988±0.0126 | **0.8195±0.0063** |
> | 1536       | 0.8029±0.0041 | **0.8310±0.0065** |
> | 2048       | 0.8057±0.0088 | **0.8394±0.0058** |
> | 2560       | 0.8150±0.0025 | **0.8404±0.0054** |
> | 3072       | 0.8175±0.0087 | **0.8441±0.0044** |
>
> AIME24：
>
> | **Budget** | **Origin**    | **TTTS**          |
> | ---------- | ------------- | ----------------- |
> | 1024       | 0.1889±0.0157 | **0.3081±0.0154** |
> | 2048       | 0.2222±0.0416 | **0.3244±0.0208** |
> | 4096       | 0.2667±0.0471 | **0.3674±0.0332** |
> | 6144       | 0.3222±0.0416 | **0.3896±0.0348** |
> | 8192       | 0.3222±0.0416 | **0.3933±0.0357** |
> | 12288      | 0.3667±0.0720 | **0.4044±0.0354** |
>
> The results show that across three runs, **TTTS consistently outperforms the original model**, demonstrating stable performance improvements.

---

> > ### Comment · Reviewer_Pihp · 2025-08-03
> >
> > Thank you for the detailed reply. I will maintain my high evaluation of the submission.

---

> ### Author Response · Authors · 2025-08-03
> **Thank you for your high evaluation of our submission!**
>
> We are grateful for your engagement in the discussion process. We are pleased that our response has adequately addressed your concerns. We will remain available to address any further questions or clarifications until the end of the discussion phase.
>
> Thank you for all the valuable comments and questions. Your comments indeed help us further improve the paper, and we will incorporate these discussions into the manuscript.
>
>
> Sincerely,
>
> The authors

---

### Official Review · Reviewer_WHTM · 2025-06-30

**Clarity:** 3
**Significance:** 3
**Originality:** 3
**Rating:** 4
**Confidence:** 4

**Summary:**

This paper discovered that reasoning-relevant tokens seemed to surge during long Chain-of-Thought (CoT) reasoning in the mutual information (MI) perspective, which is an interesting phenomenon in distilled large reasoning language models (LRMs). Through the deep analysis, the authors demonstrated that reasoning tokens played an important role in math reasoning performance: the higher MI indicated better reasoning accuracy. To validate the observation, this paper leveraged the Representation Recycling and Think Token based test-time scaling methods to intervene the LRMs inference, and results were in line with the finding.

**Questions:**

1. I notice that during the investigation, the authors conduct experiments from 7B to 70B, which is very comprehensive, while the application experiments were conducted only on 7B. I understand that there may be computation resources limited, why did not complement a 14B experiments to make the validation more convincible?
2. Intervening at the test-time has demonstrated the performance, have the authors considered for post-training phase methods? Just for personal curiosity.
3. Except for the MI, are there other alternative approaches that can support the analysis results?
4. This paper listed some frequent tokens such as: “Wait”, “Hmm” and “So”, have the authors statistical the “Alternatively” and how did it perform? As during my investigation, this word is also very common.

**Ethical Concerns:**

["NO or VERY MINOR ethics concerns only"]

**Final Justification:**

I have read all the replies from the authors, and my early minor questions and suggestions have been addressed. Hence, I maintain my positive suggestion for this submission.

**Limitations:**

The authors put the limitation section in Supplementary Materials, where they have fully discussed the current deficiencies.

**Paper Formatting Concerns:**

There is no formatting concerns of this paper.

**Quality:**

3

**Strengths And Weaknesses:**

**Strengths**
1. This work investigated the reasoning procedures in the lens of MI and observe some phenomena that correct prediction of LRMs appears with significant MI values that bind with specific tokens, which is interesting.
2. Combining extensive empirical validation and theoretical analysis in both the main body and supplementary materials, the authors revealed the MI peak patterns on both reasoning and non-reasoning LMs, they also explored the relation between token representations and MI peaks and built a set of correlations.
3. This paper further proposed two inference-time methods inspired by the former findings, and experimental results demonstrated the effectiveness of the presented methods, thereby proving the correctness of the above analysis.

**Weaknesses**
1. Although this work discovered and demonstrated the positive correlation between the MI peaks and reasoning performance from both the empirical and theoretical perspectives, the investigation and evaluation were mainly across different math problem-solving benchmarks, involving no other domains that demand reasoning, such as GPQA and MedQA, undermining the effectiveness for all reasoning tasks.
2. The token budgets of DeepSeek-R1-Distill-Qwen models were usually set the maximum as 32768, and the accuracy in Figure 7 seemed not converged, the authors might consider for more max_length to show the overall trends.
3. Generally, the evaluation of reasoning models should consider the pass@k metric with more sampling times, as the LRMs perform unstably, merely the one-time inference is not that faithful.

---

> ### Author Rebuttal · Authors · 2025-07-31
>
> Thank you for your great efforts in reviewing this paper. We will try our best to answer all your questions. **Please let us know if you still have further concerns, or if you are not satisfied with the current responses, so that we can further update the response ASAP.**
>
> ---
>
> **Q1**: "Although this work discovered and demonstrated the positive correlation between the MI peaks and reasoning performance from both the empirical and theoretical perspectives, the investigation and evaluation were mainly across different math problem-solving benchmarks, involving no other domains that demand reasoning, such as GPQA and MedQA, undermining the effectiveness for all reasoning tasks."
>
> **A1**: Thank you for your constructive comment. We have **followed your instructions to conduct new experiments on GPQA and MedQA** to examine whether the MI peaks phenomenon still exists. We used DeepSeek-R1-Distill-Llama-8B, DeepSeek-R1-Distill-Qwen-14B, and their corresponding base models.
>
> 1. **On both GPQA and MedQA, the MI peaks phenomenon persists**. Since figures cannot be included here, we instead report the AOM metric (as defined in Lines 160-162, larger values indicate stronger outlier peaks) for reference.
>
>       |                  |           | **GPQA**   | **MedQA**  |
>       | ---------------- | --------- | ---------- | ---------- |
>       | **Llama-3.1-8B** | Origin    | 3.5193     | 3.6337     |
>       |                  | Reasoning | **4.0809** | **3.8436** |
>       | **Qwen2.5-14B**  | Origin    | 2.7671     | 2.4355     |
>       |                  | Reasoning | **2.9444** | **2.6281** |
>
> 2. Regarding the **tokens at MI peaks**:
>
>    - For GPQA, the token set largely overlaps with the one obtained from the MATH dataset in the manuscript.
>
>    - For MedQA, while tokens like "Let," "So," and "But" still show high overlap, we also observe some new tokens such as "Admin," "Perform," "She," and "He," which may be related to the medical domain.
>
> In summary, these experiments further validate that the MI peaks phenomenon generalizes to other reasoning domains.
>
> ---
>
> **Q2**: "The token budgets of DeepSeek-R1-Distill-Qwen models were usually set the maximum as 32768, and the accuracy in Figure 7 seemed not converged, the authors might consider for more max_length to show the overall trends."
>
> **A2**: Thank you. We follow your suggestions to further increase the token budgets on AIME24 in Figure 7. The results show that after accuracy saturates, **TTTS consistently outperforms the baseline**.
>
> | **Budget** | **Origin** | **TTTS**  |
> | ---------- | ---------- | --------- |
> | 16384      | 33.3       | **34.87** |
> | 20480      | 33.3       | **35.55** |
> | 24576      | 33.3       | **34.87** |
> | 28672      | 33.3       | **35.88** |
>
> ---
>
> **Q3**: "Generally, the evaluation of reasoning models should consider the pass@k metric with more sampling times, as the LRMs perform unstably, merely the one-time inference is not that faithful."
>
> **A3**: Thank you for your suggestion. We have followed your suggestions to **conduct new experiments by adding pass@3 as an evaluation metric for RR and TTTS**, following settings in Lines 248-260 and 277-280. The results on AIME24 with DeepSeek-R1-Distill-Llama-8B are as follows:
>
> | **Budget** | **Origin** | **TTTS**   |
> | ---------- | ---------- | ---------- |
> | 2048       | 0.3666     | **0.3866** |
> | 3072       | 0.3666     | **0.4266** |
> | 4096       | 0.3666     | **0.4444** |
> | 6144       | 0.4333     | **0.4733** |
> | 8192       | 0.4        | **0.4888** |
>
> | **Origin** | **RR**     |
> | ---------- | ---------- |
> | 0.5        | **0.5666** |
>
> The results show that **when evaluated with pass@3, both TTTS and RR consistently outperform the baselines**.
>
> ---
>
> **Q4**: "I notice that during the investigation, the authors conduct experiments from 7B to 70B, which is very comprehensive, while the application experiments were conducted only on 7B. I understand that there may be computation resources limited, why did not complement a 14B experiments to make the validation more convincible?"
>
> **A4**: Thank you for your constructive suggestions. We have followed your suggestions to **conduct new experiments** to validate the effectiveness of RR and TTTS using **DeepSeek-R1-Distill-Qwen-14B** on AIME24 following the settings in Lines 248-260, 277-280. The following experimental results indicate that 1) **TTTS** performs worse than the origin model at early stages of the token budget but continues to grow steadily later  (similar to the phenomenon described in Lines 184–190); 2) **RR** shows a clear improvement over the origin model.
>
> | **Budget** | **Origin** | **TTTS**  |
> | ---------- | ---------- | --------- |
> | 2048       | **40.0**   | 36.91     |
> | 3072       | **43.3**   | 42.55     |
> | 4096       | 46.7       | **47.01** |
> | 6144       | 53.3       | **54.15** |
> | 8192       | 53.3       | **55.28** |
>
> | **Origin** | **RR**     |
> | ---------- | ---------- |
> | 0.5670     | **0.6670** |
>
> ---
>
> **Q5**: "Intervening at the test-time has demonstrated the performance, have the authors considered for post-training phase methods? Just for personal curiosity."
>
> **A5**: Thank you for your valuable suggestions. For simplicity and ease of use, we only applied thinking tokens during the test-time phase. Meanwhile, it can also be developed into post-training phase methods, and here are two potential integration directions:
>
> 1. During RL (Reinforcement Learning) and SFT (Supervised Fine-Tuning), higher weights can be assigned to these "thinking tokens."
> 2. During RL, multiple rollouts could be performed for thinking tokens to allow more exploration.
>
> Some more recent works are exploring similar ideas, and we are also happy to explore and verify these post-training applications in future work.
>
> ---
>
> **Q6**: "Except for the MI, are there other alternative approaches that can support the analysis results?"
>
> **A6**: A good question. Some similarity and distance measuring metrics and tools, such as Center Kernel Alignment (CKA) and Kullback–Leibler (KL) divergence, can also be used to support the analysis results. We are happy to use different metrics in future work.
>
> ---
>
> **Q7**: "This paper listed some frequent tokens such as: “Wait”, “Hmm” and “So”, have the authors statistical the “Alternatively” and how did it perform? As during my investigation, this word is also very common."
>
> **A7**: A good question.
> Yes, we did record the token “Alternatively”, which appears on the left of Figure 1(b) in the main text. Regarding its frequency, it ranks roughly within the 20th and 40th across DeepSeek-R1-Distill-Llama-8B, DeepSeek-R1-Distill-Qwen-14B, and QwQ-32B, while Figure 4 only demonstrates top-15 tokens.

---

> > ### Comment · Reviewer_WHTM · 2025-08-06
> >
> > Thank you for the responses. As the authors have addressed the questions and suggestions, I will maintain my positive rating for this submission.

---

> > > ### Author Response · Authors · 2025-08-08
> > > **Thank you for your reply!**
> > >
> > > Thank you very much for your feedback and for recognizing our efforts in the rebuttal. We are glad that our clarifications and additional results have adequately addressed your concerns.
> > >
> > > If there are any remaining issues or further clarifications that would be helpful, we would be more than happy to provide them. We will remain available to address any further questions or clarifications until the end of the discussion phase.
> > >
> > > We truly appreciate your constructive comments, which have been very valuable in improving our paper!
> > >
> > > Sincerely,
> > >
> > > The authors

---

### Official Review · Reviewer_DDE5 · 2025-07-03

**Clarity:** 2
**Significance:** 2
**Originality:** 3
**Rating:** 3
**Confidence:** 3

**Summary:**

The paper presents an information-theoretic investigation into the internal reasoning dynamics of Large Reasoning Models (LRMs). The authors take inspiration from information theory and focus on tracking the mutual information (MI) between intermediate hidden representations and the correct answer as the model generates step-by-step reasoning outputs. They claim to identify a distinctive phenomenon in LRMs, i.e., sharp MI peaks that emerge during certain reasoning steps, which are largely absent in non-reasoning LLMs. These peaks often correspond to the generation of specific tokens like "Hmm", "Wait", and "So", referred to as thinking tokens, which appear to reflect moments of self-reflection, correction, or logical transition. To leverage this observation, the authors propose two training-free inference-time techniques: Representation Recycling (RR), which reprocesses the high-MI representations within the model to enrich reasoning, and Thinking Token Test-Time Scaling (TTTS), which appends thinking tokens to prompt further reasoning and improve output quality.

The authors present empirical results showing that both RR and TTTS enhance reasoning performance across multiple math benchmarks, including GSM8K, MATH500, and AIME24. Quantitatively, they show that LRMs consistently exhibit higher MI means, greater variance, and more prominent MI peaks compared to their non-reasoning counterparts. Suppressing the generation of thinking tokens during inference leads to substantial performance degradation, while suppressing other tokens does not, confirming their functional importance. Furthermore, TTTS is shown to improve model performance even under tight token budgets, while RR offers consistent gains by reprocessing latent reasoning signals.

**Questions:**

**Questions**

* The paper defines mutual information between intermediate representations and the golden answer using ​($\hat{y}$) [specifically in Theorem 1], but it’s not clearly explained whether this includes just the answer or also the input question during computation. Since the MI is highly sensitive to what is treated as “target,” it would be good to clarify 1) Was the pre-question context included in the target representation for MI computation? 2) If not, how was temporal alignment handled between question and answer during MI estimation across timesteps? From the lines of application, this distinction is crucial for accurately interpreting the theoretical results and the empirical MI peaks as reported in the paper.

* Another question is regarding the thinking tokens; it seems many of the identified thinking tokens (e.g., “So,” “Therefore,” “Wait”) are commonly used to begin new sentences (i.e., the sentence always begins with these words/tokens). Did the authors analyze whether these tokens are simply serving as sentence-onset (SOS) markers, rather than uniquely contributing to reasoning via high-MI representations? An ablation comparing the MI of all sentence-starting tokens vs. only those labeled as thinking tokens would clarify whether their impact is truly semantic/functional or a structural artifact of generation that the Language model has learned (this is also related to the previous question, if the gold answer/target is only passed without the pre-text question, then the MI will be high due to this property of start of sentence token). A more causally grounded and controlled experiment would be required to make the proposed finding/claim more reliable and concrete.

**Minor Comments**
* The paper appears to use or refer to the "logit lens" technique, but the citation provided is incorrect or missing. Given that this is a foundational method in interpreting hidden representations, it's important to correctly acknowledge prior work (https://www.lesswrong.com/posts/AcKRB8wDpdaN6v6ru/interpreting-gpt-the-logit-lens). It is recommended to verify and correct the reference accordingly.

* Regarding the presentation quality, Figure 2 aims to show the MI evolution trajectories across different steps of the reasoning process. However, it is visually challenging to extract meaningful patterns without any prior context. The figure should better highlight the trend emphasized in Figure 1, for e.g., via vertical markers at peak positions or in-figure annotation. Additionally, the caption could explicitly state and help the reader by summarizing the key observed trends and their implications, making the figure more self-contained and easy to follow.

**Ethical Concerns:**

["NO or VERY MINOR ethics concerns only"]

**Final Justification:**

After carefully considering the rebuttal and discussion, I would maintain my overall rating. While the paper presents an interesting information-theoretic way of analyzing LLM reasoning and introduces empirical observations, several core issues remain insufficiently addressed.

* The paper introduces Theorems 1 and 2 to ground the observed MI peaks in an information-theoretic framework. However, even after the rebuttal, the theoretical framing remains oversimplified and potentially misleading, especially regarding the dependence on token length 𝑇 (the bound becoming tighter with an increase in the number of tokens seems to be counterintuitive, and may be an oversimplification). Also, as stated in the original review, the theorems restate variants of existing results (Fano-inequality bounds), without proper citations or technical advancement. This limits their originality and utility. Overall, the current theoretical component does not robustly support the primary claims and may mislead readers into overinterpreting the impact of MI accumulation or token length on LLM reasoning performance.

* The authors confirm that MI is computed using representations from an isolated answer-only pass (including reasoning steps, excluding the question). This still raises two major concerns: 1) Feeding only the answer (no question) into the model for the target representation decouples it from the context in which reasoning occurs. Even if CoT steps are included, the model isn't processing them as a response to a specific question. 2) The paper does not adequately justify why the answer is isolated in this way for MI computation. This is somewhat problematic given that reasoning is inherently context-dependent.

Though the idea of using information theory for interpretability is unique, the paper still has some open ends that were not cleared in the rebuttal phase.

**Limitations:**

Yes, the authors have acknowledged several important limitations in their analysis, particularly the granularity of reasoning steps and the lack of a mechanistic explanation for the emergence of MI peaks. However, as per the NeurIPS guidelines, it is recommended to keep them in the main paper; in the current version, they are present in the appendix.

**Paper Formatting Concerns:**

The references in the appendix appear to be maintained separately and do not integrate with the main bibliography numbering. This can confuse readers and hinder proper citation tracking. Please consider merging or cross-referencing them to ensure a consistent citation system throughout the paper.

**Quality:**

2

**Strengths And Weaknesses:**

**Strengths**

* The authors present an empirical analysis showing that MI between intermediate token representations and the final answer exhibits distinct peaks at certain “thinking tokens.” This finding (if weaknesses/questions are addressed properly) is a new information-theoretic perspective on the inner workings of language reasoning models (LRMs), showing interpretable patterns aligned with existing findings regarding procedural knowledge and the importance of reasoning steps in detail.

* The identification of tokens with high MI that correspond to linguistic markers of reasoning (e.g., “So”, “Wait”) provides an important interpretability finding that directly supports the hypothesis that LRMs engage in discrete reasoning steps (via reflection), also detectable through MI computations.

*  The paper proposes extended methods like Representation Recycling (RR) and Test-Time Thinking Token Scaling (TTTS), both training-free interventions inspired by the MI peak observations. These methods demonstrate consistent performance gains on challenging math reasoning benchmarks, showing practical value. (However, a contrast with the existing TTTS approach could be made for better highlighting the contribution, [https://arxiv.org/pdf/2501.19393v2])


**Weaknesses**

* One of the major weaknesses of this work is regarding the core theorems presented in this work. The core theorems presented (Theorems 1 and 2) essentially restate or slightly extend well-known results related to Fano’s inequality in mutual information theory. However, the paper neither cites this foundational work nor sufficiently elaborates on how its results substantively advance understanding of MI in the context of LRMs. Furthermore, Theorem 2’s bound scales with the number of tokens T, becoming tighter as T grows, a dependence that appears counterintuitive and is not adequately justified, raising concerns about the practical significance and interpretation of the theoretical analysis. A more detailed justification for increasing the context length and a suitable relation to Fano’s inequality would be good to add in the updated version of the paper.

* The Test-Time Thinking Token Scaling approach strongly resembles the “Simple Test-Time Scaling” technique from existing literature (s1: Simple test-time scaling). The paper could contrasts with this prior work (as mentioned in one of the inspirations [line 270]), highlighting the originality and potential impact of the TTTS method. [https://arxiv.org/pdf/2501.19393v2]

* There are some ambiguities in the MI calculation setup, particularly it is not clearly specified whether the “golden answer” representation used for computing MI includes the pre-question context or only the answer itself. This lack of clarity affects reproducibility and the interpretation of the MI peak dynamics, making it hard to understand in one go and keeping things ambiguous in general.

* Regarding the analysis of “thinking tokens”, the paper does not provide a clear, explicit identification method of the tokens termed as “thinking tokens.” (seems more like a heuristic). Moreover, many of these tokens seem like the sentence-initial markers, which could confound the interpretation that MI peaks correspond to reasoning steps rather than simply sentence boundaries. A systematic analysis of the overlap between sentence-start tokens and thinking tokens is missing, making it more difficult to come to a proper conclusion/finding about the thinking tokens.

---

> ### Author Rebuttal · Authors · 2025-07-31
>
> Thank you for your great efforts in reviewing this paper. We will try our best to answer all your questions. **Please let us know if you still have further concerns, or if you are not satisfied with the current responses, so that we can further update the response ASAP.**
>
> ---
>
> **Q1:** "One of the major weaknesses of this work is regarding the core theorems presented in this work. The core theorems presented (Theorems 1 and 2) essentially restate or slightly extend well-known results related to Fano’s inequality in mutual information theory. However, the paper neither cites this foundational work nor sufficiently elaborates on how its results substantively advance understanding of MI in the context of LRMs. Furthermore, Theorem 2’s bound scales with the number of tokens T, becoming tighter as T grows, a dependence that appears counterintuitive and is not adequately justified, raising concerns about the practical significance and interpretation of the theoretical analysis. A more detailed justification for increasing the context length and a suitable relation to Fano’s inequality would be good to add in the updated version of the paper."
>
> **A1**: Thank you for your valuable comments. We will follow your suggestions to add more connections with Fano's inequality and cite relevant works, so as to make the positioning of the theoretical part clearer.
>
> - **In Lines 138-141 and 144-151, we discuss how the theoretical part helps us understand the MI peaks phenomenon.** Specifically, the higher the accumulated MI during the reasoning process, the lower the probability of the LLMs making mistakes. The emergence of MI peaks will potentially increase the final accumulated MI. Experimental results in Figure 3 and Table 2 verify that LRMs have emerged with the phenomenon of MI peaks, and improve the overall MI during the reasoning process (Lines 171-179) compared with their base models. At the same time, the reasoning performance of LRMs is also significantly better than that of base models.
> - **Explaining the number of tokens T in Theorem 2**. Intuitively, as T increases, allowing more MI to be accumulated enables the LLM to explore more during the reasoning stage to approach the correct answer, thus reducing the lower bound of the model's error probability. In addition, the experimental results in Figure 7 in the main text verify the theoretical results: as the number of tokens T increases, the model's performance improves (i.e., the lower and upper bounds of errors in Theorems 2 & 3 are getting tighter, respectively).
>
> ---
>
> **Q2**: "The Test-Time Thinking Token Scaling approach strongly resembles the “Simple Test-Time Scaling” technique from existing literature (s1: Simple test-time scaling). The paper could contrasts with this prior work (as mentioned in one of the inspirations [line 270]), highlighting the originality and potential impact of the TTTS method."
>
> **A2**: Thank you. The prior work "s1: Simple test-time scaling" has already been cited and discussed in Lines 267-271. The main innovation and contribution of this paper is to analyze the underlying dynamic reasoning mechanisms of LRMs, discovering the MI Peaks phenomenon, and further analyzing the token distribution corresponding to MI Peaks based on information-theoretic tools. Furthermore, to demonstrate the potential real-world application of MI Peaks' theoretical insights, we show two case studies (RR and TTTS) for improving reasoning performance in a training-free manner. In this way, this paper aims to demystify and explain the reasoning mechanisms of LRMs rather than merely focusing on Test-time scaling.
>
> ---
>
> **Q3**: "There are some ambiguities in the MI calculation setup, particularly it is not clearly specified whether the “golden answer” representation used for computing MI includes the pre-question context or only the answer itself. This lack of clarity affects reproducibility and the interpretation of the MI peak dynamics, making it hard to understand in one go and keeping things ambiguous in general." "... Was the pre-question context included in the target representation for MI computation? ... "
>
> **A3**: Thank you.  As introduced in Lines 90-91, we describe that "extract the representation of the gold answer by feeding y into the LLM." In this way, the representation used for computing MI only includes the answer itself.
>
> ---
>
> **Q4**: "Regarding the analysis of “thinking tokens”, the paper does not provide a clear, explicit identification method of the tokens termed as “thinking tokens.” (seems more like a heuristic). Moreover, many of these tokens seem like the sentence-initial markers, which could confound the interpretation that MI peaks correspond to reasoning steps rather than simply sentence boundaries. A systematic analysis of the overlap between sentence-start tokens and thinking tokens is missing, making it more difficult to come to a proper conclusion/finding about the thinking tokens." "... how was temporal alignment handled between question and answer during MI estimation across timesteps? ..." "... Another question is regarding the thinking tokens; it seems many of the identified thinking tokens (e.g., “So,” “Therefore,” “Wait”) are commonly used to begin new sentences (i.e., the sentence always begins with these words/tokens). Did the authors analyze whether these tokens are simply serving as sentence-onset (SOS) markers, rather than uniquely contributing to reasoning via high-MI representations? ..."
>
> **A4**:  Thank you for your suggestions. As introduced in Lines 51-53 and 212-216, we define the tokens at MI peaks as thinking tokens.
>
> We have **followed your suggestion and extracted the words at the beginning of all sentences**, then sorted them by their occurrence frequency to get the top ten: "So, Wait, Let, But, Therefore, The, Alternatively, Then, Hmm, First". Interestingly, we found that there is a significant overlap between these sentence-onset (SOS) tokens and the thinking tokens at MI peaks in Figure 4. Here, we would like to make the following two statements:
>
> 1. **The large overlap between the set of tokens at MI peaks and the set of SOS tokens is not surprising**. The beginning of a sentence usually affects the overall direction of the subsequent content, so it may contain more information, which precisely corresponds to those MI peaks. From another perspective, $I(X;Y) = H(X) - H(X|Y)$, where $H(X)$ measures the uncertainty of $X$. The tokens at the beginning of a sentence usually affect the subsequent trend and require transitions, choices, etc., thus having higher uncertainty; while for tokens in the middle of a sentence, since the semantics of the previous part of the sentence have gradually been determined, it is only necessary to continue generating along the established logic/semantics, thus having higher certainty (this point has also been mentioned in some concurrent works [c1]).
> 2. **Considering the frequency of token occurrence further, the distribution of tokens at MI peaks and the distribution of SOS tokens are quite different**. This also indicates that not all SOS tokens have high mutual information.
>
> [c1] Beyond the 80/20 Rule: High-Entropy Minority Tokens Drive Effective Reinforcement Learning for LLM Reasoning
>
> ---
>
> **Q5: Minor Comments**
>
> **A5**: Thank you for your constructive suggestions. We will follow your suggestions to make the following revisions:
>
> 1. Cite and discuss the "logit lens" technique.
> 2. Add more notations in Figure 2 to highlight the MI peaks phenomenon, and enrich the captions with descriptions and brief conclusions, so that each figure is more self-contained.
> 3. Move the limitation section into the main paper.

---

> > ### Comment · Reviewer_DDE5 · 2025-08-06
> >
> > Thank you for the detailed rebuttal. Some of the concerns still remain open, and further explanation would make this discussion better:
> >
> > * Regarding Theorem 2 and bound with token length, while I understand the motivation to interpret longer contexts, allows greater cumulative MI and potentially lower error rates, I guess the current framing may not be adequate and can lead to confusion or overinterpretation (if I understood correctly). Specifically, the tightening of the error bound with increased token length T seems to suggest that LLM can keep reducing prediction error simply by increasing the number of reasoning steps. This feels counterintuitive for two primary reasons: 1) In real settings, there is a natural upper bound to the total information that can be extracted from the input. Simply increasing the number of reasoning steps cannot keep injecting more signal into the process. Yet, the bound as framed does not clearly reflect any such saturation. 2) Also, empirically, reasoning performance does not improve indefinitely with longer sequences. After a point, LLMs may even reduce in performance due to error accumulation, distractions, or loss of coherence. Thus, a theoretical bound that becomes tighter (i.e., implies less error) as T increases indefinitely without modeling diminishing returns may oversimplify the practical behavior of reasoning in LLMs.
> >
> > * Regarding MI computation setup and representation of the “golden answer, " thanks for confirming that the MI is computed solely based on the answer token, excluding the pre-question context. This raises a few more clarifications that are open 1) it would be good to confirm what representation is being used for the answer-only sequence, are you using the final-layer hidden states from the LLM when it is fed just the answer tokens (i.e., the embedding after forward-passing the answer tokens alone), or are you using the static embedding vectors from the embedding matrix? If the answer is being processed without any surrounding context (no input question or reasoning steps preceding it), then many layers in the transformer might behave similarly across examples, and the resulting hidden states might not vary significantly or contain strong task-relevant information. It’s not immediately clear what the benefit of isolating the answer in this way is, given that in practice, the answer is almost always conditioned on the question and intermediate reasoning. Hence, it would help to explain the rationale behind excluding the question context and to consider whether context-conditioned answer representations might offer a more meaningful target for MI estimation. Otherwise, there's a risk that the MI signal is partly driven by artifacts of how "unconditioned" the answer is, rather than its informativeness to reasoning.
> >
> > * Regarding the thinking tokens vs. sentence-onset tokens, thank you for conducting the frequency analysis of sentence-onset tokens and comparing them with thinking tokens. However, the finding of substantial overlap also shows a possibility of a confounder that would need closer inspection. It would be good to perform a more quantitative overlap analysis using metrics such as Jaccard similarity between the set of thinking tokens and the set of top sentence-start tokens. This will help clarify whether these are largely the same set or only partially overlapping. Overall, if the overlap between SOS tokens and thinking tokens turns out to be very high (e.g., IoU > 0.5), it may require more clarification on the interpretation that thinking tokens represent moments of logical transition or reflection, rather than simply serving as structural boundaries that naturally carry higher uncertainty.

---

> > > ### Author Response · Authors · 2025-08-08
> > > **Response to Reviewer DDE5 (Part 1/3)**
> > >
> > > We are grateful for your engagement in the discussion process! Thank you again for your great efforts in reviewing this paper. We will try our best to answer all your remaining questions.
> > >
> > > **Please let us know if you have further questions, or if you are not satisfied with the current responses. We will remain available to address any further questions or clarifications until the end of the discussion phase.**
> > >
> > > ---
> > >
> > > **Q1**: "Regarding Theorem 2 and bound with token length, while I understand the motivation to interpret longer contexts, allows greater cumulative MI and potentially lower error rates, I guess the current framing may not be adequate and can lead to confusion or overinterpretation (if I understood correctly). Specifically, the tightening of the error bound with increased token length T seems to suggest that LLM can keep reducing prediction error simply by increasing the number of reasoning steps. This feels counterintuitive for two primary reasons: 1) In real settings, there is a natural upper bound to the total information that can be extracted from the input. Simply increasing the number of reasoning steps cannot keep injecting more signal into the process. Yet, the bound as framed does not clearly reflect any such saturation. 2) Also, empirically, reasoning performance does not improve indefinitely with longer sequences. After a point, LLMs may even reduce in performance due to error accumulation, distractions, or loss of coherence. Thus, a theoretical bound that becomes tighter (i.e., implies less error) as T increases indefinitely without modeling diminishing returns may oversimplify the practical behavior of reasoning in LLMs."
> > >
> > > **A1**: Thank you for your constructive comment.
> > >
> > > 1. In this manuscript, Theorems 1 & 2 are intended to provide insights and explanations for the **MI peaks phenomenon** (see Lines 127–132), rather than for the length of generated answers. In this way, Theorems 1 & 2 suggest that **when the token length $T$ is fixed**, **the presence of MI peaks may lead to a higher cumulative MI**, and thus potentially a lower error probability (Lines 147–151). This implication is also partially supported by the experimental results in Figure 3 (Lines 176–179). Therefore, Theorems 1 & 2 help build a theoretical connection between the MI peaks phenomenon and reasoning performance.
> > > 2. We also **fully agree with your concerns** about potential misunderstandings when interpreting the theorems under the increased token length $T$. If these theorems are used to further understand LRMs' reasoning behavior, your points are highly valid. We are more than happy to **follow your suggestions and improve the theoretical part** by:
> > >    -  Introducing a lemma to state the information upper bound: $\sum_{j}^{T}I(y;h_j | h_{<j}) \leq C$, where $C$ is a constant. This reflects your point that *"there is a natural upper bound to the total information that can be extracted from the input."* The proof sketch is: first we have $\sum_{j}^{T}I(y;h_j | h_{<j}) = I(y;h_{1:T}) \leq \min \\{H(y), H(h_{1:T}) \\}$; since in practical settings the representations can be viewed as discrete variables, so the entropy is bounded, and thus  $\min \\{H(y), H(h_{1:T})\\} < C$ holds.
> > >    -  Add Eq. (2) with an additional term $+f(T, \sigma)$, where $\sigma$ denotes the potential noise during each step reasoning.  This reflects your point that *"LLMs may even reduce in performance due to error accumulation."*  The function $f$ can model a variety of relationships, such as super-linear relationship [c1],  approximately linear relationship [c2], and other more complex relationships [c3-c4].
> > >
> > > In summary, Theorems 1 & 2 already help to establish a theoretical link between the **MI peaks phenomenon** and model reasoning performance. However, we fully agree that they are not sufficient for fully understanding LRMs' reasoning. We are willing to **follow your suggestions and incorporate the above theoretical improvements and additional discussion to further strengthen the theoretical section**.
> > >
> > > [c1] Rethinking External Slow-Thinking: From Snowball Errors to Probability of Correct Reasoning
> > >
> > > [c2] When More is Less: Understanding Chain-of-Thought Length in LLMs
> > >
> > > [c3] Beyond Exponential Decay: Rethinking Error Accumulation in Large Language Models
> > >
> > > [c4] Reasoning Can Hurt the Inductive Abilities of Large Language Models

---

> > > ### Author Response · Authors · 2025-08-08
> > > **Response to Reviewer DDE5 (Part 2/3)**
> > >
> > > **Q2**: "Regarding MI computation setup and representation of the “golden answer, " thanks for confirming that the MI is computed solely based on the answer token, excluding the pre-question context. This raises a few more clarifications that are open 1) it would be good to confirm what representation is being used for the answer-only sequence, are you using the final-layer hidden states from the LLM when it is fed just the answer tokens (i.e., the embedding after forward-passing the answer tokens alone), or are you using the static embedding vectors from the embedding matrix? If the answer is being processed without any surrounding context (no input question or reasoning steps preceding it), then many layers in the transformer might behave similarly across examples, and the resulting hidden states might not vary significantly or contain strong task-relevant information. It’s not immediately clear what the benefit of isolating the answer in this way is, given that in practice, the answer is almost always conditioned on the question and intermediate reasoning. Hence, it would help to explain the rationale behind excluding the question context and to consider whether context-conditioned answer representations might offer a more meaningful target for MI estimation. Otherwise, there's a risk that the MI signal is partly driven by artifacts of how "unconditioned" the answer is, rather than its informativeness to reasoning."
> > >
> > > **A2**: Thank you for your comment.
> > >
> > > 1. We would first like to clarify a potential misunderstanding: although the ground-truth answers do not include the question, they do **include the** **full chain-of-thought reasoning steps** **[c1-c4]**, rather than just a single final answer token. These answers are then fed into the LLM to obtain the final-layer hidden states, **not** static embedding vectors from the embedding matrix. As a result, each ground-truth representation does carry **task-relevant information related to the specific question**.
> > >
> > > 2. In addition, we have **followed your suggestion to conduct new experiments** **by prepending the question to the answer**. In this setting, each answer includes the question, reasoning steps, and final result. The experimental results are summarized below:
> > >
> > >   -  The **MI peaks phenomenon still persists**. Since figures cannot be shown here, we report the mean, std, and AOM metrics (as defined in Lines 160–162, larger AOM indicates stronger outlier peaks), following **Table 2** in the manuscript. The AOM trends closely match those in Table 2. We will also include these MI peaks visualizations (similar to Figure 2) in the revised manuscript.
> > >
> > >        |      | **Llama-3.1-8B** |  |      **Qwen2.5-14B**  |           |
> > >         | ---- | ---------------- | --------------- | ------ | --------- |
> > >         |      | Origin           | Reasoning       | Origin | Reasoning |
> > >         | Mean | 0.0900           | 0.1163          | 1.3127 | 3.2922    |
> > >         | Std  | 0.0583           | 0.0680          | 0.4094 | 0.6632    |
> > >         | AOM  | 3.3649           | 4.3745          | 2.4463 | 2.9313    |
> > >
> > >    -  The **token distribution at MI peaks remains almost unchanged**. Below are the top-10 tokens at MI peaks for each model, showing strong consistency with Figure 4 in the manuscript:
> > >
> > >         - *DeepSeek-R1-Distill-Llama-8B*: So, Let, The, Hmm, I, That, Since, Then, To, But
> > >         - *DeepSeek-R1-Distill-Qwen-14B*: So, Let, Hmm, I, Okay, That, First, Now, But, Wait
> > >
> > > We hope the above clarification and additional experiments could help validate the reasonableness of our experimental design and the generality of the MI peaks phenomenon.
> > >
> > > [c1] Measuring Mathematical Problem Solving With the MATH Dataset
> > >
> > > [c2] Training Verifiers to Solve Math Word Problems
> > >
> > > [c3] NuminaMath: The largest public dataset in AI4Maths with 860k pairs of competition math problems and solutions
> > >
> > > [c4] Let's Verify Step by Step

---

> > > ### Author Response · Authors · 2025-08-08
> > > **Response to Reviewer DDE5 (Part 3/3)**
> > >
> > > **Q3**: "Regarding the thinking tokens vs. sentence-onset tokens, thank you for conducting the frequency analysis of sentence-onset tokens and comparing them with thinking tokens. However, the finding of substantial overlap also shows a possibility of a confounder that would need closer inspection. It would be good to perform a more quantitative overlap analysis using metrics such as Jaccard similarity between the set of thinking tokens and the set of top sentence-start tokens. This will help clarify whether these are largely the same set or only partially overlapping. Overall, if the overlap between SOS tokens and thinking tokens turns out to be very high (e.g., IoU > 0.5), it may require more clarification on the interpretation that thinking tokens represent moments of logical transition or reflection, rather than simply serving as structural boundaries that naturally carry higher uncertainty."
> > >
> > > **A3**: Thank you for your comments.
> > >
> > > 1. We have **followed your suggestion to compute the Jaccard similarity** between the sets of thinking tokens and sentence-onset tokens. Additionally, considering the importance of token frequency rankings, we also include the **Rank-Biased Overlap** **(RBO)** metric, which measures the similarity between two ranked lists. RBO ranges from 0 to 1, with higher values indicating greater overlap and stronger agreement in ordering. The results are as follows:
> > >
> > >     |                              | **Jaccard Similarity** | **RBO** |
> > >       | ---------------------------- | ---------------------- | ------- |
> > >       | DeepSeek-R1-Distill-Qwen-7B  | 0.3333                 | 0.3888  |
> > >       | DeepSeek-R1-Distill-Llama-8B | 0.4286                 | 0.3645  |
> > >       | DeepSeek-R1-Distill-Qwen-14B | 0.4286                 | 0.3982  |
> > >
> > >    The result show that, despite some overlap between thinking tokens and sentence-onset tokens, both the **Jaccard similarity and RBO are below 0.5**, suggesting differences in both token sets and their frequency ranks.
> > >
> > > 2. Furthermore, **structural boundaries do not always naturally carry higher MI with the answer** (from another perspective, higher uncertainty). Specifically, we compute the MI between the representations of three types of tokens and the answer. The results are summarized below:
> > >
> > >     |                              | **Thinking tokens** | **Sentence-onset tokens** | **All tokens** |
> > >       | ---------------------------- | ------------------- | ------------------------- | -------------- |
> > >       | DeepSeek-R1-Distill-Qwen-7B  | 5.2344              | 3.7956                    | 3.3016         |
> > >       | DeepSeek-R1-Distill-Llama-8B | 0.3958              | 0.3238                    | 0.1279         |
> > >       | DeepSeek-R1-Distill-Qwen-14B | 5.6198              | 4.0788                    | 3.3508         |
> > >
> > > 3. We would like to summarize the key logical relationship as follows:
> > >
> > > - We first observe the MI peaks phenomenon, and then decode the representations at those MI peaks into token space, and find that these tokens often express logical transitions or reflections. We call these tokens "thinking tokens".
> > > - Then following your suggestions, we find that 30–50% of these thinking tokens also appear as top sentence-onset tokens. We believe this overlap is reasonable: as mentioned in our first-round rebuttal, sentence beginnings often influence the direction of the following content, and thus may carry more information (with higher uncertainty), which corresponds to those MI peaks.
> > > - However, the reverse is not necessarily true: **not all sentence-onset tokens exhibit high MI with the answer** (with high uncertainty).
> > >
> > > We hope the above explanations and additional experimental results could help clarify the distinctions and connections between thinking tokens and sentence-onset tokens.

---

### Official Review · Reviewer_xCDc · 2025-07-04

**Clarity:** 3
**Significance:** 3
**Originality:** 4
**Rating:** 4
**Confidence:** 4

**Summary:**

This paper investigates the reasoning dynamics of large reasoning models from an information-theoretic perspective, focusing on how mutual information (MI) between intermediate representations and the correct answer evolves during the reasoning process. The authors observe a phenomenon they refer to as "MI peaks," where certain tokens exhibit sharp increases in MI during reasoning. These tokens often correspond to reflective or transitional expressions like "Hmm," "Wait," and "Therefore," which the authors term "thinking tokens." The paper further demonstrates that suppressing these thinking tokens significantly harms reasoning performance, while other tokens have minimal impact. Based on these insights, two methods, Representation Recycling (RR) and Thinking Token-based Test-time Scaling (TTTS), are proposed to improve reasoning performance. Experimental results show consistent improvements across multiple benchmarks.

**Questions:**

Please refer to the weakness

**Ethical Concerns:**

["NO or VERY MINOR ethics concerns only"]

**Final Justification:**

I thank the authors for their thorough rebuttal and the additional experimental results. They have convincingly addressed most of my initial concerns, reinforcing my confidence in the paper's merits. Consequently, I will maintain my positive score and recommend the paper for acceptance.

**Limitations:**

Yes

**Quality:**

4

**Strengths And Weaknesses:**

### Strength
1. The paper presents a well-structured and methodical approach to analyzing reasoning dynamics in LRMs. It starts with identifying MI peaks, then explores their semantic meaning, validates their functional importance, and finally proposes practical applications. The flow is intuitive and easy to follow, making the findings accessible even to readers not deeply familiar with information theory.

2. The experimental results consistently support the theoretical claims throughout the paper. Both proposed methods (RR and TTTS) demonstrate stable improvements in reasoning performance across different datasets (e.g., AIME24, MATH500, GSM8K). These results suggest that the insights gained from the MI peak analysis are not only theoretically sound but also practically useful for enhancing LRM performance.

### Weakness
1. While the paper identifies MI peaks as important phenomena, it does not address why MI values across different models (even of the same family) are not directly comparable. Additionally, although model sizes vary (e.g., 7B, 8B, 14B, etc.), no clear relationship between model size and MI behavior is established. Given that larger models generally exhibit better reasoning capabilities, the lack of correlation between MI and model scale raises questions about the universality of the MI peak phenomenon.

2. The experiment comparing the suppression of thinking tokens versus randomly selected non-thinking tokens may not be entirely fair. Since non-thinking tokens include many function words or punctuation marks that naturally carry less semantic content, randomly suppressing them might disproportionately affect meaningless tokens. A more rigorous and informative evaluation necessitates the careful selection of token pools that are both semantically meaningful and sufficiently challenging for comparative analysis.

3. The RR method, introduced in Lines 240–248, lacks sufficient detail and hard to understand. Specifically, it is unclear whether this mechanism is equivalent to prompting the model to “think again” using the same input—a strategy akin to asking the model to generate multiple responses and choosing the best one. To validate RR’s novelty and effectiveness, the authors should compare it against a baseline where the model generates multiple responses without recycling and evaluate whether the improvement stems from the recycling mechanism itself or simply from ensemble-like effects.

4. In evaluating TTTS, the paper assumes that appending thinking tokens improves reasoning by encouraging deeper reflection. However, the improvement could also stem from merely extending the output length. A more rigorous evaluation would involve testing whether appending other non-thinking but semantically neutral continuation tokens (e.g., “continue,” “more”) yields similar benefits. Without such a control, it is hard to determine whether the improvement is specifically due to the semantic role of thinking tokens or just the increased token budget.

---

> ### Author Rebuttal · Authors · 2025-07-31
>
> Thank you for your great efforts in reviewing this paper. We will try our best to answer all your questions. **Please let us know if you still have further concerns, or if you are not satisfied with the current responses, so that we can further update the response ASAP.**
>
> ---
>
> **Q1**: "While the paper identifies MI peaks as important phenomena, it does not address why MI values across different models (even of the same family) are not directly comparable. Additionally, although model sizes vary (e.g., 7B, 8B, 14B, etc.), no clear relationship between model size and MI behavior is established. Given that larger models generally exhibit better reasoning capabilities, the lack of correlation between MI and model scale raises questions about the universality of the MI peak phenomenon."
>
> **A1**: This is a good question.
>
> - **There is no clear relationship between model size and MI behaviors according to experimental results in Tables 1 and 2 of the main text**. Specifically, quantitative metrics including #MI Peaks, #All Steps, Ratio of MI Peaks, and Max Interval do not show obvious patterns as the model size scales. Different families and sizes of LLMs have different architectures and dimensions of latent representations, and are trained with different strategies and data. These factors are all closely related to the reasoning capabilities, which need to be further investigated in the community [c1-4].
> - Nevertheless, **MI peak is a universal phenomenon across LLMs' families (Llama and Qwen) and sizes (7B，8B，14B，32B, and 70B)**, as shown in Figures 2-3 in the main text, Section 2, and Figures 2-13 in Appendix.
>
> [c1] Demystifying Long Chain-of-Thought Reasoning in LLMs
>
> [c2] SimpleRL-Zoo: Investigating and Taming Zero Reinforcement Learning for Open Base Models in the Wild
>
> [c3] LIMO: Less is More for Reasoning
>
> [c4] Dr.GRPO Understanding R1-Zero-Like Training: A Critical Perspective
>
> ---
>
> **Q2**: "The experiment comparing the suppression of thinking tokens versus randomly selected non-thinking tokens may not be entirely fair. Since non-thinking tokens include many function words or punctuation marks that naturally carry less semantic content, randomly suppressing them might disproportionately affect meaningless tokens. A more rigorous and informative evaluation necessitates the careful selection of token pools that are both semantically meaningful and sufficiently challenging for comparative analysis."
>
> **A2**: Thank you for your comment.
>
> - We would like to first clarify that in the original experiments, when selecting random tokens, we filtered out punctuation and other meaningless symbols, keeping only English words.
>
> - In addition, we have **followed your instructions to select more meaningful tokens for comparison**. Specifically, using the training set of the MATH dataset, we extract tokens from model answers by (i) removing the thinking tokens, (ii) requiring English tokens with length > 2, and (iii) selecting the top-frequency tokens. These tokens include *answer, final, can, which, from, have, etc*. Based on this, we follow the experimental setting of Figure 5 and set the number of suppression tokens to 15.
>
>   - | **Dataset**              | **GSM8K** | **MATH500** | **AIME24** |
>     | ------------------------ | --------- | ----------- | ---------- |
>     | **Random**               | 82.9      | 79.2        | 33.3       |
>     | **Thinking tokens**      | 69.0      | 43.8        | 20.0       |
>     | **Top-frequency tokens** | 81.3      | 69.8        | 26.7       |
>
>   -  The results show that suppressing **top-frequency tokens** hurts model performance more than suppressing random tokens. However, suppressing **thinking tokens** still **leads to the most significant performance drop**.
>
> ---
>
> **Q3:** "The RR method, introduced in Lines 240–248, lacks sufficient detail and hard to understand. Specifically, it is unclear whether this mechanism is equivalent to prompting the model to “think again” using the same input—a strategy akin to asking the model to generate multiple responses and choosing the best one. To validate RR’s novelty and effectiveness, the authors should compare it against a baseline where the model generates multiple responses without recycling and evaluate whether the improvement stems from the recycling mechanism itself or simply from ensemble-like effects."
>
> **A3**: Thank you for your valuable suggestions.
>
> - **Intuitively, the core idea of RR design is that when the LLM aims to generate "thinking tokens", we do not let LLM directly generate subsequent tokens. Instead, we feed the representation of the current token back into the same layer once more**, as introduced in Lines 244-245, hoping that the LLM can better utilize the information in the representation of the "thinking token". For example, suppose the sequence currently generated by the LLM is "... Step 4 is finished. " At this point, we detect that the next token going to be generated is "So" (one of "thinking tokens" with high MI). Instead of letting the LLM generate "So" directly, we send the corresponding representation of this token back to the current layer once again. Then, the LLM may realize that it needs to check the previous results, thus generating "Wait". The subsequent generation process then returns to normal, unless a "thinking token" is encountered again, in which case the above process is repeated. **Therefore, RR is not equivalent to prompting the LLM to "think again" using the same input**.
>
> - Prompting the LLM to "think again" means asking the LLM to re-answer the question, while RR only reuses the representations of some key steps to "temporarily" intervene in its thinking process. In this way, the former requires far more computing resources than RR, so a direct comparison is unfair. In fact, prompting the LLMs to "think again" and RR are two **orthogonal methods that can be used in combination**. To verify this, **we conducted new experiments** using DeepSeek-R1-Distill-Llama-8B on AIME24.
>
>   - | Method             | Result |
>     | ------------------ | ------ |
>     | Original           | 33.0   |
>     | "think again"      | 50.0    |
>     | RR                 | 40.0    |
>     | "think again" + RR | 56.66 |
>
> - The experimental results show that both "think again" and RR can improve the performance, but their working principles are different ("think again" requires significantly more computing resources). Moreover, **"think again" can be used in combination with RR to further improve the model's performance**.
>
> ---
>
> **Q4**: "In evaluating TTTS, the paper assumes that appending thinking tokens improves reasoning by encouraging deeper reflection. However, the improvement could also stem from merely extending the output length. A more rigorous evaluation would involve testing whether appending other non-thinking but semantically neutral continuation tokens (e.g., “continue,” “more”) yields similar benefits. Without such a control, it is hard to determine whether the improvement is specifically due to the semantic role of thinking tokens or just the increased token budget."
>
> **A4**: Thanks for your comments.
>
> - We agree with your point that "merely extending the output length can improve model performance." We already mentioned this in Lines 267–271, and the performance of the "Origin" line in Figure 7 (where only the token budget is increased) also confirms this observation.
>
> - In addition, we **have followed your suggestion to conduct new** **continuous generation experiments** using neutral tokens such as "continue" and "more."
>
>   - | **Budget** | **Origin** | **TTTS** | **Continue&More** |
>     | ---------- | ---------- | -------- | ----------------- |
>     | 512        | 57.2       | 59.83    | 59.9              |
>     | 1024       | 63.2       | 65.65    | 65.2              |
>     | 2048       | 70.4       | 72.02    | 70.5              |
>     | 3072       | 73.2       | 75.35    | 72.9              |
>     | 4096       | 75         | 77.27    | 75.4              |
>
>   -  The results show that **continuation with neutral tokens like "continue" and "more" is less effective than continuation with thinking tokens**.

---

> > ### Comment · Reviewer_xCDc · 2025-08-06
> >
> > Thank you for the detailed rebuttal and the additional experimental results. They have successfully addressed most of my concerns. This has strengthened my confidence in the paper's contributions. Therefore, I will maintain my positive score.

---

> > > ### Author Response · Authors · 2025-08-08
> > > **Thank you for your reply!**
> > >
> > > Thank you very much for your feedback and for recognizing our efforts in the rebuttal. We are glad that our clarifications and additional results have addressed your concerns and strengthened your confidence in our work.
> > >
> > > If there are any remaining issues or further clarifications that would be helpful, we would be more than happy to provide them. We will remain available to address any further questions or clarifications until the end of the discussion phase.
> > >
> > > We truly appreciate your constructive comments, which have been very valuable in improving our paper!
> > >
> > > Sincerely,
> > >
> > > The authors

---

### Decision · Program_Chairs · 2025-09-17

**Decision:**

Accept (poster)

**Comment:**

This paper presents an interesting analysis showing that there are critical tokens in thinking based LLMs where the mutual information (MI) between the token representation and the answer representation spikes – these are identified as “thinking tokens”. On further analysis, the top thinking tokens look like “Hmm”, “Wait” and “Therefore,” that express reflection or transition and are shown to be crucial for the LLM’s performance since suppressing them leads to large drops in accuracy. These findings inspire some inference-time ideas that practically improves the accuracy of LLMs by looping on representations of thinking tokens and inserting more thinking tokens.

---

The reviewers liked the overall idea and execution of the paper. There were some concerns about baselines, evaluation metrics and the significance of the theoretical result. The author's response addressed most of the concerns.

R1: MI is not comparable across scales, better baseline for token suppression, different baselines for the proposed inference-time methods —- *Addressed by showing MI spikes at many scales and adding more baselines*

R2: Theory follows from Fano’s inequality, possibly misrepresentative and loose, issues with how MI was calculated —- *Partially addressed through relaxed theory and more details on MI calculation*

R3: Not enough benchmarks. Larger token budget and pass@k —- *Addressed by adding 14B results and pass@k evaluations*

R4: More details on what MI signifies, results follows from Fano —- *Promised to address*


The theoretical results, as multiple reviewers pointed out, basically follow from known results. So that’s not the most significant contribution of the paper but simply justifies the existence of such critical tokens with large MI.

---


Overall the paper makes an interesting observation and shows some practical utility for it. The authors also did a fair job of addressing most reviewer concerns. On the whole, the paper makes a positive contribution towards better understanding the thinking mechanism in LLMs. Thus the recommendation is accept, assuming the authors incorporate the reviewer suggestions (especially about the use of Fano’s inequality in the theory)